# Efficient Multi Subject Visual Reconstruction from fMRI Using Aligned Representations

## Abstract

Reconstructing visual images from fMRI data is a central but highly challenging problem in neuroscience. Despite recent progress, current methods fall short when data and computation are limited——precisely the conditions under which this task is most critical. We introduce a novel architecture-agnostic training paradigm to improve fMRI-based visual reconstruction through a subject-agnostic common representation space. We show that it is possible to leverage subject-specific lightweight modules to develop a representation space where different subjects not only lie in a shared space but are also aligned semantically. Our results demonstrate that such a training pipeline achieves significant performance gains in low-data scenarios. We supplement this method with a novel algorithm to select the most representative subset of images for a new subject. Using both techniques together, one can fine-tune with at most 40% of the data while outperforming the baseline trained with the minimum standard dataset size. Our method generalizes across different training paradigms and architectures, producing state-of-the-art performance and demonstrating that a subject-agnostic aligned representation space is the next step towards efficient Brain-Computer Interfaces.

## 1 Introduction

Over the past several decades, the use of machine learning techniques has enabled the decoding and/or reconstruction of information represented in neural activation patterns. These approaches offer a key step toward characterizing and quantifying cognition Naselaris et al. (2011). Beyond establishing the presence of information in brain activity, another important goal is to compare decoding or reconstruction performance across experimental manipulations, such as cued visual attention (Serences and Boynton, 2007; Kamitani and Tong, 2005; Scolari et al., 2012; Sprague and Serences, 2013; Itthipuripat et al., 2019; Sprague et al., 2018) and visual working memory (Serences et al., 2009; Harrison and Tong, 2009; Ester et al., 2013; Sprague et al., 2014; Christophel et al., 2012; Li et al., 2021). However, for these approaches to be practical in a cognitive neuroscience laboratory, it is typically necessary to acquire large amounts of data.

Recent efforts have begun to address the challenges associated with decoding and reconstructing brain activity. Early approaches, including self-supervised learning frameworks (Beliy et al., 2019; Gaziv et al., 2022; Mozafari et al., 2020; Shen et al., 2019; Seeliger et al., 2018; Ren et al., 2021) struggled to generate semantically accurate image reconstructions, often resulting in images with limited resemblance to the original stimuli. Significant advancements were achieved by incorporating multimodal data, such as text (Lin et al., 2022), and then by leveraging the generative power of latent diffusion models (LDMs) (Takagi and Nishimoto, 2023; Scotti et al., 2023; Lu et al., 2023; Xia et al., 2024). A more detailed literature review is presented in Appendix A. However, these state-of-the-art methods require extremely large datasets with dozens of hours of fMRI data per participant. Thus, there is a clear need to maximize the efficiency and generalizability of image reconstruction methods so that a model for a new participant can be trained using as little data as possible.

Recent works by Scotti et al. (2024) and Liu et al. (2024) have attempted to improve generalization by projecting fMRI signals into a shared representation space across subjects. However, in these pipelines the shared space is treated only as a convergence point of subject-specific pathways and is not explicitly aligned. We show in our analysis that though subject representations in this space share a similar structure, manipulations such as a simple orthogonal transformation substantially

increase cross-subject similarity. This motivates an explicit alignment strategy: instead of relying on the shared space to emerge implicitly, lightweight subject-specific neural network modules called adapters (Liu et al., 2024) can be trained to align each subject to a reference subject.

We propose Adapter Alignment (AA), an efficient and generalizable approach to fMRI-based visual reconstruction using a subject-agnostic representation space. Instead of fine-tuning new subjects end-to-end, we align new subjects to a pre-trained reference subject using adapter modules. AA substantially reduces data and training time while improving reconstruction quality. We also present a data-efficient image selection strategy that identifies the most representative training samples by covering key dimensions of variance in the common space. This method reduces the burden of data collection and further complements Adapter Alignment by making fine-tuning more efficient. Our contributions are as follows:

- We demonstrate that fMRI signals from different subjects exhibit structural similarity in a shared representation space. While orthogonal transformations improve alignment, near-perfect cross-subject alignment can be attained using adapter modules (Section 2).
- We introduce Adapter Alignment (AA), a novel training paradigm where new subjects are explicitly aligned to a reference subject. AA achieves faster convergence and better performance, especially in low-data regimes (Sections 3.2, 4.1 and 4.2).
- We propose a submodular greedy algorithm to identify representative training images, providing a data-efficient alternative to random sampling. This strategy consistently improves fine-tuning performance across subjects and complements Adapter Alignment by reducing the amount of data required. (Sections 3.4 and 4.2)
- We validate AA on both the NSD and THINGS (Hebart et al., 2023) datasets, showing generalization across different architectures (Section 4.4), data acquisition protocols (7T vs. 3T) (Section 5), and subject pools (Appendix D).

## 2 A COMMON VISUAL REPRESENTATION SPACE

At the core of our approach is the hypothesis that each individual's brain (or at least the tissue involved in processing static visual images) can be viewed as an instance of a shared 'common brain', with visual activity patterns that vary along similar dimensions in a shared representational space. This idea is supported by prior neuroscience findings, including consistent category-selective tissue responses (e.g., faces, scenes) and continuous dimensions of high-level visual selectivity in ventral temporal cortex such as animate/inanimate, large/small (real-world size), and spiky/round (Huth et al., 2012; Long et al., 2018; Bao et al., 2020). Retinotopic organization provides another example, where adaptive models transform individual anatomical mappings from retinal to cortical coordinates into a shared reference space (Sprague and Serences, 2013; Wandell and Winawer, 2015). In our work, we analyze embeddings from multiple subjects and provide evidence that although fMRI signals do not naturally align in a shared space, they exhibit a similar underlying structure. We further explore structural and semantic properties of this space through eigenvector analysis, visual embedding comparisons, and alignment experiments, with additional results of semantic categories (e.g. animate / inanimate) provided in Appendix K.

Existing works, such as MindEye2 (Scotti et al., 2024), employ simple linear adapters to project subject-specific fMRI signals into a common representation space. A key indication of the existence of such a space is precisely this structural similarity across subjects. To investigate it, we use a pre-trained MindEye2 model to extract common-space embeddings for shared images across multiple subjects. Initial cross-subject cosine similarity is low, but as shown in Figure 1, applying an orthogonal transformation with Procrustes analysis (Schönemann, 1966) reveals that the embeddings can be structurally aligned. However, this alignment is still incomplete, reflecting both the limitations of orthogonal transformations and the absence of an explicit alignment objective.

As a preliminary experiment, we pre-trained a single-subject model and explicitly mapped subsequent subjects to this reference subject using lightweight subject-specific adapters. Even with a simple adapter design (a linear layer with a GELU), this approach yielded near-perfect alignment in the common space (Figure 1 (c)). The choice of adapter is not critical. Any transformation that goes beyond an orthogonal mapping can substantially improve alignment highlighting that it is the explicit alignment objective, rather than architectural complexity, that drives the improvement.

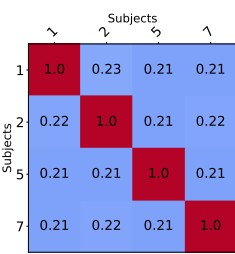
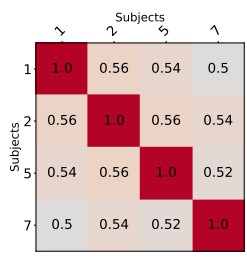
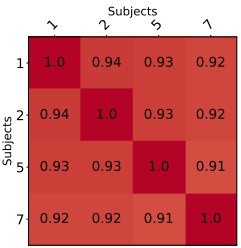

(a) Original alignment.    (b) Post orthogonal transformation.    (c) Post Adapter Alignment.

Figure 1: Cross-subject cosine similarity of shared image embeddings in the common space extracted from a pre-trained MindEye2 model. (a) Alignment between subjects is weak before any transformation. (b) Applying an orthogonal transformation substantially improves cross-subject similarity. (c) Explicit alignment through lightweight adapter modules achieves near-perfect alignment. Only Subjects 1, 2, 5, and 7 are shown for consistency throughout the manuscript.

The above evidence of structurally well-aligned representation spaces across subjects forms the core hypothesis of our work. In the remainder of this paper, we develop a training paradigm that explicitly enforces such alignment and evaluate it empirically against the commonly used baseline fine-tuning approach. Additional analysis is provided in Appendix J.

## 3 METHODS

In the previous section, we established the existence of a shared representation space across subjects. We now describe the methods used to leverage this space for fMRI-to-image reconstruction. Our framework introduces Adapter Alignment (AA) as a lightweight strategy for subject-specific alignment, integrates this into a reconstruction pipeline, and augments it with a greedy image selection algorithm to further improve data efficiency. Together, these components form the basis for the experiments presented in the following section.

### 3.1 DATASET

We use the Natural Scenes Dataset (NSD) (Allen et al., 2022) but apply a different split compared to the traditional setup. Typically, common images across subjects are assigned to the test set, while the training set consists of unique images viewed by each subject. However, for effective Adapter Alignment, it is critical to have a one-to-one mapping of images across subjects. To achieve this, we incorporate the common images into the training set and swap an equal amount of training images to the test sets. This means that every subject now has a unique set of images on which they are tested. The test split is chosen to roughly ensure an equal distribution across different categories in the dataset. We present the performance of the original training method and the proposed technique on the new split to ensure a fair comparison across subjects and alignment techniques. Three splits are generated in this manner and whenever results are presented on the new split, it is an average of the performance on all 3 splits.

Our initial experiments revealed that models performed slightly worse on the unique image test set than on the common image test set, suggesting that the unique image test set presents a more challenging benchmark. Similar to prior work, we focus on Subjects 1, 2, 5, and 7, as they have complete 40-hour fMRI datasets available in NSD, enabling better reconstruction performance. These subjects provide robust data for training and evaluation, allowing us to validate our alignment and reconstruction methods comprehensively.

### 3.2 ADAPTER ALIGNMENT

Current approaches for training multi-subject pipelines with a common representation space typically follow an end-to-end strategy, where fMRI inputs are processed by the model and the loss is computed at the final layer to minimize the discrepancy between the ground-truth and predicted embeddings.

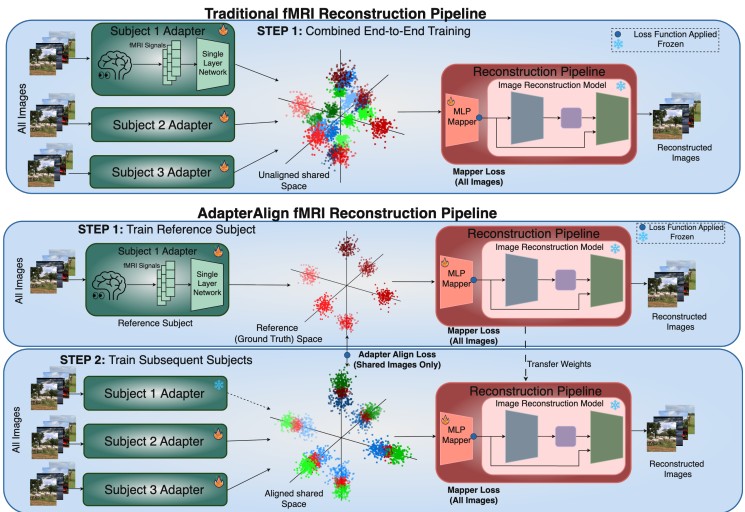

Figure 2: Training procedures for fMRI-Image Reconstruction. **Top:** Traditional Approach to reconstruction. All subjects are trained simultaneously and loss functions are applied at the MLP Mapper to align the output to pre-trained CLIP space. **Bottom:** Proposed AdapterAlign training pipeline. First, we train a single reference subject and obtain its embeddings in the shared space. Then we train subsequent subjects to minimize loss at both the shared space and the mapper. The same mapper weights are transferred over in step 2.

Throughout the paper, we refer to this fine-tuning strategy as the baseline. In this setup, the common space emerges organically during training without explicit constraints. However, this approach has two key drawbacks: (1) it requires extended training to form a satisfactory common space and achieve convergence, and (2) the subject-specific embeddings in the common space are not strongly aligned, making it harder for the residual MLP to map multiple subject inputs to a unified output (Figure 2).

To address these challenges, we propose Adapter Alignment (AA) training. In AA training, the pipeline is first pre-trained end-to-end on a single reference subject, chosen based on reconstruction performance and data availability. Among NSD subjects, Subject 1 consistently exhibited the best reconstruction performance, making it the ideal candidate to construct the reference space. After pre-training, the output embeddings for the subject-specific adapter, particularly those corresponding to the shared images, are extracted.

When fine-tuning a new subject, we adopt a two-step process. The first is **Adapter Alignment**, where we train the new subject's adapter using the shared images, with the objective of aligning its output to the embeddings of Subject 1 for the same images. The adapter is allowed to overfit on these shared images until the loss is minimized. The second step is **End-to-End Fine-tuning**, where we unfreeze the entire pipeline and resume training, applying the usual losses at the final output. Additionally, we introduce an MSE loss at the adapter level specifically for the common images, which allows the adapter to restructure itself to account for subject-specific discrepancies and adapt to the unique images encountered by the new subject. We refer to this process of overfitting the adapter with AA and fine-tuning end-to-end "AAMax". From our preliminary experiments, we found that AAMax outperforms other methods of aligning adapters like the ones proposed by Ferrante et al. (2024) and Wang et al. (2024) Both methods do not fine-tune the MLP and the diffusion prior end-to-end after aligning the adapters which reduces the generalization ability of the fine-tuned models. We compare the performance between these training methods in Appendix F. We find that AAMax is the best method for fine-tuning new subjects in limited data settings.

This training method is effective across both high-data and low-data regimes. With full training data, AAMax achieves performance comparable to traditional end-to-end training while converging more quickly. In limited data settings, AAMax shows a clear advantage, producing higher-quality reconstructions and greater accuracy with substantially fewer training samples. We outline our method in Figure 2 and detail each component of the pipeline in the following section.

## 3.3 Reconstruction Pipeline

In general, fMRI-to-image reconstruction pipelines follow a common structure: brain signals are mapped into an intermediate embedding space (semantic or virtual), which is then decoded into an image using a pre-trained generative model such as a diffusion decoder. We discuss multiple previous approaches and their pipelines in Appendix A. Since Adapter Alignment (AA) is agnostic to architectural choices, we adopt the pipeline used in MindEye1 for our main experiments and refer readers to MindEye1 (Scotti et al., 2023) and MindEye2 (Scotti et al., 2024) for architectural details. We also validate our method on MindEye2's architecture (Section 4.4 and Appendix B) and across varying component choices and shared space dimensionalities (Appendix I), showing consistent improvements and compatibility.

In summary, flattened fMRI voxels from the "nsdgeneral" region are passed through a subject-specific adapter (a single linear layer with GELU non-linearity) into a shared representation space. This shared representation is then processed by a residual MLP. The resulting output is mapped into the CLIP (Radford et al., 2021) space via a pre-trained diffusion prior (Ramesh et al., 2022). This serves as a conditioning signal for a diffusion model to produce the final reconstruction.

We detail the training losses used in this pipeline for both the high-level (semantic) and low-level (visual) branches below, with full equations provided in Appendices N and O. The overall loss function for the high-level pipeline is:

$$\mathcal{L}_{\text{new}} = \begin{cases} \lambda_1 \mathcal{L}_{\text{Prior}} + \lambda_2 \mathcal{L}_{\text{CLIP}}, & \text{for unique images} \\ \lambda_1 \mathcal{L}_{\text{Prior}} + \lambda_2 \mathcal{L}_{\text{CLIP}} + \lambda_3 \mathcal{L}_{\text{MSE}}(\mathbf{z}_{\text{new}}, \mathbf{z}_{\text{ref}}), & \text{for common images} \end{cases}$$

where $\mathbf{z}_{\text{adapter, new}}$ and $\mathbf{z}_{\text{adapter, ref}}$ are the adapter-level embeddings for the new and reference subjects, respectively, on the shared common images.

The low-level pipeline also follows a similar structure to the high-level pipeline but maps the fMRI data to latent embeddings derived from a custom autoencoder. This autoencoder is trained on downscaled images from the NSD dataset (64x64x3) and reduces them to a latent space of size 1024. The low-level pipeline learns to map to this latent space; passing these embeddings through a pre-trained decoder produces blurry reconstructions. The loss function for this stage is:

$$\mathcal{L}_{\text{new}} = \begin{cases} \mathcal{L}_{\text{MSE}}(\mathbf{z}_{\text{fMRI, new}}, \mathbf{z}_{\text{AE}}), & \text{for unique images} \\ \mathcal{L}_{\text{MSE}}(\mathbf{z}_{\text{fMRI, new}}, \mathbf{z}_{\text{AE}}) + \mathcal{L}_{\text{MSE}}(\mathbf{z}_{\text{adapt, new}}, \mathbf{z}_{\text{adapt, ref}}), & \text{for common images} \end{cases}$$

where $\mathbf{z}_{\text{fMRI, new}}$ is the output embedding for the new subject, $\mathbf{z}_{\text{AE}}$ is the AE latent space representation of the image and $\mathbf{z}_{\text{adapter, new}}$ and $\mathbf{z}_{\text{adapter, ref}}$ are the adapter-level embeddings for the new and reference subjects, respectively.

After mapping brain signals to the CLIP space, the mappings are fed as conditioning to a frozen Diffusion Model (Xu et al., 2023) to generate final reconstructions.

## 3.4 Greedy Algorithm for Best Candidate Image Selection

In the context of fMRI-based visual reconstruction, we introduce a greedy algorithm that selects the most representative subset of images from the common space. We first decompose (SVD) the adapter's weight matrix of the reference subject. Then we project the embeddings of the shared set of 1000 images onto the $d$ principal dimensions corresponding to the principal singular vectors. For each dimension $j$, we partition the range into $B_j$ bins that is determined by the ratio of singular value $\lambda_j$ to the largest singular value $\lambda_1$:

$$B_j = \left\lfloor w \cdot \frac{\lambda_j}{\lambda_1} \right\rfloor, \quad j = 1, 2, \ldots, d$$

where $w$ is a predefined scaling parameter. Dimensions with larger singular values are partitioned into more bins, reflecting their greater variance and importance in capturing meaningful data features.

Each image (and its fMRI) maps to exactly 1 bin in each dimension and our goal is to find the smallest subset of images that covers all bins in each dimension. This subset is critical in selecting the most representative images for a new subject, ensuring that the selected images capture the key activity

patterns that generalize across subjects. The problem as stated is NP-hard. However, it is submodular and a greedy heuristic such as given below achieves an $(1 - 1/e)$ approximation ratio.

Given a subset $S$ of images, let $Gap(S, j)$ be the number of empty bins in dimension $j$, and let $Gap(S) = \Sigma_j Gap(S, j)$ be the total number of empty bins across all dimensions. At every step, the greedy algorithm chooses image $i \notin S$ that minimizes $Gap$ when added to $S$.

After applying the image selection algorithm, the chosen subset simply replaces random sampling for fine-tuning. The training procedure itself is unchanged: the selected images can be used with either baseline fine-tuning or AAMax. In this way, the algorithm serves purely as a strategy to identify the most representative subset of images, reducing the data requirement without altering the reconstruction pipeline. Formal proofs for the algorithm are presented in Appendix L.

## 4 RESULTS

We now evaluate the performance of our proposed framework across a range of experimental conditions. We begin with complete data experiments, where AAMax is compared against baseline training. We then turn to limited data settings, where we report results from both AAMax and AAMax combined with image selection in a unified comparison. We also analyze how AAMax scales with increasing amounts of training data, highlighting that its advantage diminishes as larger datasets allow even unaligned models to generalize. Finally, we present experiments showing that AAMax generalizes across architectures using MindEye1 and MindEye2 (Scotti et al., 2024) architectures.

### 4.1 TESTS ON COMPLETE DATA

We first validate our training method using the full dataset. Subject 1 is used as the reference subject, while fine-tuning is performed on Subjects 2, 5, and 7. The results averaged over all 3 subjects are presented in Table 1. AA is agnostic to the choice of reference subject but we found that having a good reference subject improves fine-tuning performance (Appendix D). We use the same set of metrics as Scotti et al. (2024) to evaluate our reconstructions. Pixel Correlation, SSIM and 2-way percent correct for the 2nd and 5th layer of AlexNet are considered low level metrics. 2-way percent correct for Inception(Szegedy et al., 2016), CLIP, EfficientNet(Tan, 2019) and SwAV(Caron et al., 2020) score are considered high level metrics. When trained to completion, both baseline fine-tuning and AAMax show similar overall performance. However, the MSE loss at the adapter level is significantly lower (by an order of magnitude as shown in Appendix J.2) when using adapter alignment. This allows for more precise post-hoc analysis after extracting embeddings from the common space.

Table 1: Quantitative results on training the reconstruction models on all 40 hours of data. The fine-tuning results are averaged over subjects 2,5 and 7. AAMax and baseline fine-tuning perform similarly across all metrics.

| Method | Low-Level | | | | High-Level | | | |
| | PixCorr ↑ | SSIM ↑ | Alex(2) ↑ | Alex(5) ↑ | Incep ↑ | CLIP ↑ | Eff ↓ | SwAV ↓ |
| --- | --- | --- | --- | --- | --- | --- | --- | --- |
| Subject 1 Pre-train | 0.345 | 0.346 | 92.80% | 96.88% | 94.40% | 90.02% | 0.692 | 0.399 |
| Baseline Fine-Tune AVG | 0.258 | **0.339** | **89.64%** | 95.05% | 92.63% | **89.64%** | 0.717 | 0.427 |
| AAMax Fine-Tune AVG | **0.259** | 0.337 | 89.20% | **95.09%** | **92.68%** | 89.49% | **0.713** | **0.423** |

### 4.2 TESTS ON LIMITED DATA

A key objective of multi-subject pipelines is to achieve accurate models of new subjects with limited data. Because acquiring fMRI data is costly, reducing the data requirements makes reconstruction pipelines more practical and broadly applicable.

In our experiments, we designate Subject 1 as the reference and fine-tune Subjects 2, 5, and 7 using the shared images. For consistency, we use the same reference subject checkpoint across all experiments and report averaged results over three runs with fixed random splits. Table 2 presents the unified results for fine-tuning with 1 hour of data (250 images), comparing baseline training and AAMax with both random sampling and image selection (IS). Across all subjects, AAMax

significantly outperforms the baseline, while IS consistently improves performance regardless of the fine-tuning strategy. As expected, the strongest results are obtained when combining AAMax with IS, showing that the two approaches are complementary. For IS, we select images based on the top 20 eigenvector dimensions, with ablations on this choice provided in Appendix E. We run experiments with scaling the amount of training data in the following section and additional ablations with 2 and 4 hours of data are included in Appendix G, where the same trend holds. We also run experiments on convergence in Appendix G and find that AA followed by just 1 epoch of End-to-End fine-tuning is good enough to outperform the baseline.

We further investigate the efficiency of our approach by testing how far data requirements can be reduced while maintaining reconstruction performance. In this setup, Subjects 2 and 5 were fine-tuned from Subject 1 using 100, 150, and 200 training images. At 100 images, both AAMax and AAMax+IS already match the baseline trained with 1 hour of data, and at 150 images they clearly surpass it. As shown in Figure 3, AAMax+IS consistently achieves the strongest results, though even AAMax alone outperforms the baseline in these low-data regimes. These findings highlight the robustness of adapter alignment and demonstrate that effective reconstructions can be achieved with as little as 40% of the data previously considered necessary.

Table 2: Limited data fine-tuning results (1 hour, 250 images) using Subject 1 as the reference and Subjects 2, 5, and 7 as fine-tuning subjects. Each subject is evaluated under four settings: Baseline with random sampling, AAMax with random sampling, Baseline with image selection (IS), and AAMax with IS. Results are averaged over three runs with fixed random splits. AAMax consistently outperforms the baseline, IS provides additional gains in both training regimes, and the best performance is achieved by combining AAMax with IS. Pixel-level metrics are shown in gray since they are not optimized in these runs but are still valid.

| Method | Low-Level | | | | High-Level | | | |
|---|---|---|---|---|---|---|---|---|
| | PixCorr ↑ | SSIM ↑ | Alex(2) ↑ | Alex(5) ↑ | Incep ↑ | CLIP ↑ | Eff ↓ | SwAV ↓ |
| Subject 2 FT 1HR Baseline | 0.078±0.010 | 0.264±0.007 | 75.33%±1.62% | 84.18%±1.40% | 74.15%±0.99% | 73.66%±1.3% | 0.862±0.011 | 0.524±0.009 |
| Subject 2 FT Selected 1HR Baseline | 0.097±0.003 | 0.288±0.004 | 78.78%±0.41% | 85.78%±0.69% | 77.13%±0.76% | 76.17%±0.03% | 0.842±0.003 | 0.513±0.005 |
| Subject 2 FT 1HR AAMax | 0.112±0.005 | 0.259±0.025 | 78.75%±1.29% | 88.04%±1.19% | 78.99%±0.97% | 78.28%±0.39% | 0.824±0.003 | 0.513±0.014 |
| Subject 2 FT Selected 1HR AAMax | 0.102±0.003 | 0.267±0.11 | **81.41%±0.27%** | **89.55%±0.26%** | **80.43%±0.13%** | **80.33%±0.01%** | **0.823±0.002** | **0.496±0.002** |
| Subject 5 FT 1HR Baseline | 0.082 | 0.267 | 75.05% | 83.28% | 76.04% | 76.40% | 0.855 | 0.526 |
| Subject 5 FT Selected 1HR Baseline | 0.098 | 0.286 | 77.07% | 86.31% | 78.09% | 79.55% | 0.827 | 0.499 |
| Subject 5 FT 1HR AAMax | 0.105 | 0.278 | 79.90% | 88.68% | 80.63% | 82.20% | 0.808 | 0.493 |
| Subject 5 FT Selected 1HR AAMax | 0.117 | 0.285 | **81.05%** | **89.88%** | **81.92%** | **82.47%** | **0.799** | **0.480** |
| Subject 7 FT 1HR Baseline | 0.079 | 0.252 | 72.98% | 79.98% | 71.57% | 72.07% | 0.871 | 0.539 |
| Subject 7 FT Selected 1HR Baseline | 0.099 | 0.286 | 78.41% | 84.77% | 74.91% | 75.79% | 0.844 | 0.515 |
| Subject 7 FT 1HR AAMax | 0.104 | 0.285 | 80.04% | 87.47% | 79.17% | 78.93% | 0.826 | 0.509 |
| Subject 7 FT Selected 1HR AAMax | 0.123 | 0.298 | **81.24%** | **88.34%** | **79.94%** | **79.81%** | **0.820** | **0.495** |

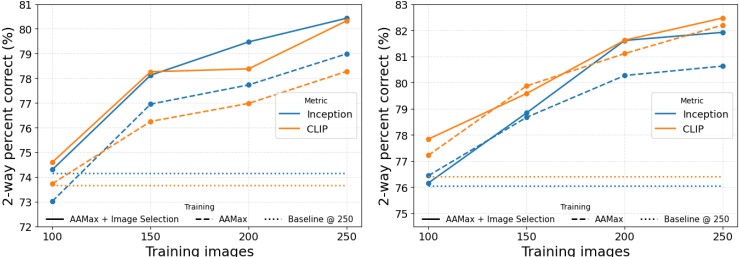

Figure 3: Pushing the limits of data efficiency with AAMax+Image Selection. Left: Subject 2 fine-tuned from Subject 1. Right: Subject 5 fine-tuned from Subject 1. In both cases, AAMax and AAMax+Image Selection outperform the baseline trained with 250 images, even when using only 150 images (a 40% reduction) and match performance with 100 images (a 60% reduction).

### 4.3 Scaling with Training Data

We further examine how the benefits of Adapter Alignment scale with training data size. Using Subject 2 as the fine-tuning subject, we progressively increase the training set from 250 to 500, 1000, and 4000 images, before extending to the full dataset. Up to 1000 images, only common images are used for alignment and fine-tuning; for 4000 images, alignment is still performed on the 1000

shared images, while the additional 3000 images are used only for fine-tuning. As shown in Figure 4, AAMax achieves substantially higher performance than baseline fine-tuning in low-data regimes, with the gap narrowing as training data increases. At full data, both approaches converge, consistent with prior multi-subject pipelines such as MindEye2, where large amounts of subject-specific data allow the model to generalize even without explicit alignment.

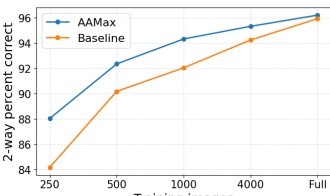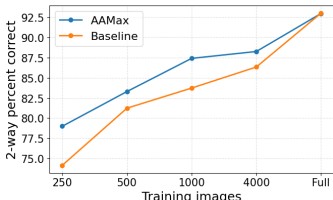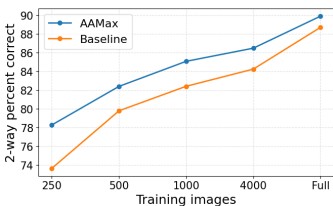

Figure 4: Performance of AAMax compared to baseline fine-tuning as a function of training data size. Left: AlexNet-5 2-way percent correct. Middle: Inception 2-way percent correct. Right: CLIP 2-way percent correct. Across all three metrics, AAMax achieves higher accuracy in limited-data regimes, with the advantage diminishing as more data becomes available.

## 4.4 GENERALIZING TO DIFFERENT ARCHITECTURES

Adapter Alignment is inherently architecture-agnostic: any reconstruction model can be extended with subject-specific adapters, enabling alignment to a common space. For efficiency and comparability with prior work, all of our main experiments are conducted using a slightly modified MindEye1 pipeline. To demonstrate the architecture-agnostic nature of Adapter Alignment, we additionally evaluate a limited-data setting using 1 hour of training data to fine-tune Subject 2. We compare baseline fine-tuning and AAMax under both the MindEye1 and MindEye2 architectures, and observe consistent improvements with AAMax across both models. Additional experiments using MindEye2 are presented in Appendix B.

Table 3: Evaluating Adapter Alignment across architectures. Subject 1 is trained as the reference, and Subject 2 is fine-tuned with 1 hour of training data. Results are shown for MindEye1 and MindEye2 under the baseline and AAMax fine-tuning. These results demonstrate that AA is architecture-agnostic. Moreover, AA scales with the underlying architecture—stronger models such as MindEye2 achieve higher absolute performance, with their gains further enhanced by AA.

| | Low-Level | | | | High-Level | | | |
|---|---|---|---|---|---|---|---|---|
| Method | PixCorr ↑ | SSIM ↑ | Alex(2) ↑ | Alex(5) ↑ | Incep ↑ | CLIP ↑ | Eff ↓ | SwAV ↓ |
| MindEye1 Subject 2 Baseline | 0.078 | 0.264 | 75.33% | 84.18% | 74.15% | 73.66% | 0.862 | 0.524 |
| MindEye1 Subject 2 AAMax | 0.103 | 0.283 | 81.77% | 89.39% | 80.31% | 78.71% | 0.824 | 0.495 |
| MindEye2 Subject 2 Baseline | 0.104 | 0.354 | 81.71% | 88.06% | 78.84% | 74.15% | 0.845 | 0.499 |
| MindEye2 Subject 2 AAMax | 0.140 | 0.355 | **82.22%** | **89.67%** | **82.97%** | **78.79%** | **0.821** | **0.483** |

## 5 GENERALIZING TO OTHER DATASETS

To demonstrate that our proposed method generalizes across datasets, we conducted experiments on the THINGS dataset. We used a model pre-trained on Subject 1 from the NSD dataset and fine-tuned it on Subject 1 from the THINGS dataset. Since a one-to-one mapping does not exist between the THINGS and NSD datasets, we build an implicit common set by comparing the train images of THINGS with the NSD common set to find a subset of semantically similar image pairs. CLIP is used to identify semantically similar images. In the end, 150 images are chosen as the common set for the THINGS dataset. As shown in Table 4, AAMax again outperforms conventional fine-tuning. This result is especially impressive as AAMax was able to semantically align the images even without an explicit alignment. We present additional experiments in Appendix J to show that AAMax actually improves alignment in the shared space.

Figure 5: Examples of reconstructed images from Subject 1 of the THINGS dataset after fine-tuning. Each pair depicts the ground truth (left) and the reconstructed image (right).

Several factors may explain the observed performance differences between the two datasets. While a plurality of recent fMRI reconstruction reports have focused on analyses of the large and high-quality Natural Scenes Dataset, which measured high-SNR 7 Tesla fMRI data at a 1.8 mm resolution, such data is often out of reach for most neuroscience labs. The modal imaging setup available to many cognitive neuroscience labs is a 3T scanner, such as a Siemens Prisma, which can achieve relatively high-quality images at 2–2.5 mm resolution and reasonable sampling rate (0.5–2 s). The THINGS-fMRI dataset was recently made available, which includes 3T fMRI data acquired at 2 mm resolution (with multiband acceleration) while participants viewed images drawn from the THINGS object database. While, in principle, this dataset is very similar to the NSD, there are notable differences in SNR (7T vs 3T), stimulus presentation protocol (NSD: 3 s stimulus presentation/1 s ITI; THINGS: 0.5 s stimulus presentation/4 s Inter-Trial Interval (ITI)), stimulus set (NSD: MS-COCO, THINGS: curated object images), and task performed by participants (NSD: recognition memory on each image; THINGS: object or not?). Thus, differences in decoding or reconstruction performance are expected, and not themselves a demonstration that models do not reliably generalize across datasets.

Table 4: Quantitative results on fine-tuning THINGS-fMRI Subject 1 using NSD Subject 1 as the reference subject. Only the high level pipeline is trained and therefore, pixelwise metrics are excluded. AAMax outperforms the baseline even when an explicit data mapping is not available.

| | Low-Level | | High-Level | | | |
|---|---|---|---|---|---|---|
| Method | Alex(2) ↑ | Alex(5) ↑ | Incep ↑ | CLIP ↑ | Eff ↓ | SwAV ↓ |
| NSD Subj1 Pre-train | 92.80% | 96.88% | 94.40% | 90.02% | 0.692 | 0.399 |
| THINGS Subj1 Individual | 68.02% | 72.6% | 58.8% | 64.29% | 0.947 | 0.649 |
| THINGS Subj1 Baseline Fine-Tune | 77.46% | 81.07% | 71.06% | 73.16% | 0.911 | 0.619 |
| THINGS Subj1 AAMax Fine-Tune | **78.27%** | **85.55%** | **74.10%** | **81.37%** | **0.902** | **0.595** |

## 6 CONCLUSIONS & DISCUSSION

In this work, we provide evidence that fMRI signals from different subjects can be structurally aligned within a shared representation space, enabling subject-agnostic modeling of visual brain activity. Building on this, we introduce Adapter Alignment (AA), a novel training paradigm that consistently outperforms traditional end-to-end pipelines in low-data regimes while achieving comparable performance in high-data settings. We further propose a greedy image selection algorithm that identifies representative training images, together reducing the data required for fine-tuning by up to 60% while matching baseline performance, and by 40% while significantly outperforming it. These contributions demonstrate that accurate visual reconstructions can be achieved with substantially less data than previously thought feasible, greatly improving the efficiency of fMRI-based reconstruction.

Our experiments show that AA generalizes across subjects, architectures, and datasets, highlighting its robustness and broad applicability. By leveraging the structure of the shared representation space, we make fMRI-based visual reconstruction more efficient and accessible. Future work could investigate richer adapter designs and extend these alignment strategies to other brain signal modalities, further advancing the goal of generalizable brain-to-image reconstruction.

## 7 REPRODUCIBILITY STATEMENT

To ensure reproducibility, we provide an anonymized codebase with all hyperparameters specified, along with clear instructions for running the experiments and clean notebooks covering training, fine-tuning, reconstruction, and evaluation. All our models were trained for 200-300 epochs on a batch size of 16, with a variable learning rate from 3e-4 to 6e-5. All the experiments were run on a single NVIDIA H200 GPU with 141GB of RAM. We build upon existing open-source repositories (MindEye1 and MindEye2) and used publicly available datasets: the NSD and THINGS datasets.

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

APPENDIX CONTENTS

## A RELATED WORK

Recent works have achieved impressive results by mapping fMRI data to latent diffusion model (LDM) spaces (Takagi and Nishimoto, 2023; Scotti et al., 2023; Lu et al., 2023; Xia et al., 2024), while simultaneously integrating multiple modalities. Despite this progress, these methods have not been thoroughly tested for their generalization performance across a larger population of subjects. In other words, constructing a model for a new subject still requires retraining the model from scratch to ensure optimal performance.

Approaches like Ferrante et al. (2024) have demonstrated cross-subject brain decoding using limited data samples by aligning multiple subjects. Although the authors, like us, leveraged visual stimuli commonly used for multiple subjects, the fidelity of their reconstructed images lags behind current advancements. Scotti et al. (2024), produced significant results in image reconstruction under limited data conditions but did not fully exploit the properties of the common representation space. In contrast, our novel approach enables faster convergence and improved performance in a limited data setting by effectively leveraging the structure of the common space. Wang et al. (2024) introduced a cross-subject brain-decoding framework that unifies fMRI dimensions through adaptive aggregation and learns subject-invariant representations via cyclic reconstruction. This resembles the second stage of our proposed strategy, yet as we show, an explicit alignment followed by end-to-end fine-tuning yields superior performance. Finally, Dai et al. (2025) manages to align subjects directly in voxel space using categorical priors and many-to-many mappings, motivated by the absence of shared stimuli, whereas our strategy operates within a learned common representation space with or without explicit mapping, generalizing effectively even when training data are weaker or heterogeneous.

Qian et al. (2023) refers to a 'common space'; however, their representation is primarily anatomical and only implicitly semantic. Thual et al. (2022) focuses on aligning the entire cortical surfaces of different individuals in a way that preserves both anatomical structure and functional activation patterns. However many downstream tasks like mapping fMRI signals to semantic embeddings for image reconstruction benefit from representations that capture high-level semantic content. Therefore the suggested approach doesn't necessarily capture abstract, semantic features (such as

object categories, scenes, or other high-level concepts) that are crucial for decoding complex stimuli. Thual et al. (2023) is an extension of Thual et al. (2022). They apply functional alignment on top of the anatomical alignment. The method is very efficient even in limited data settings but it's less integrated, meaning that it appears more as a preprocessing step. Piecewise shared response modeling Bazeille et al. (2021) provides significant performance in terms of aligning neural data from different subjects. However, the independent alignment of segments might not integrate seamlessly into a coherent global representation. Similarly to MindEye2, but in the context of M/EEG data, Li et al. (2024) manage to reconstruct visual stimuli using cross subject decoding. They tackle inter-subject variability primarily by incorporating subject-specific tokens into their M/EEG encoder.

## B  FINE-TUNING ON DIFFERENT SUBJECTS WITH MINDEYE2

The main text presents results on fine-tuning Subject 2 using Subject 1 as reference. Here we present the results for Subject 5 and Subject 7 too with 1 hour of training data. In both cases, AAMax outperforms the baseline.

Table 5: Fine-tuning on Subjects 5 and 7 using MindEye2's architecture. AAMax outperforms the baseline. The same reference subject checkpoint was used in all the runs and all models were trained with the exact same hyperparameter setup.

| | Low-Level | | | | High-Level | | | |
|---|---|---|---|---|---|---|---|---|
| Method | PixCorr ↑ | SSIM ↑ | Alex(2) ↑ | Alex(5) ↑ | Incep ↑ | CLIP ↑ | Eff ↓ | SwAV ↓ |
| Subject 5 FT 1HR Baseline | 0.102 | **0.358** | 78.40% | 86.61% | 79.85% | 77.23% | 0.839 | 0.494 |
| Subject 5 FT 1HR AAMax | **0.121** | 0.354 | **80.68%** | **88.86%** | **85.12%** | **81.89%** | **0.799** | **0.469** |
| Subject 7 FT 1HR Baseline | 0.103 | 0.356 | 78.50% | 84.70% | 76.81% | 74.02% | 0.858 | 0.506 |
| Subject 7 FT 1HR AAMax | **0.131** | **0.390** | **80.80%** | **87.86%** | **80.29%** | **76.55%** | **0.836** | 0.508 |

## C  GENERALIZING TO DIFFERENT ADAPTER ARCHITECTURES

To test the sensitivity of Adapter Alignment to the choice of adapter, we compared several lightweight variants. As shown in Table 6, the choice of adding or removing a non-linearity, or substituting different activation functions (e.g., GELU vs. ReLU), does not significantly affect alignment or reconstruction performance. In practice, simpler architectures tend to perform slightly better, with a single-layer adapter outperforming deeper variants. Our adapter choice in the main experiments was guided by the underlying backbone: we use a nonlinear (linear + GELU) adapter with MindEye1 and retain the linear adapter design of MindEye2 for consistency with the original paper. Overall, these results indicate that the specific adapter design does not affect the validity of Adapter Alignment, and performance is robust across lightweight variants.

Table 6: Ablation on adapter variants for Subject 2 fine-tuned with 1 hour of data under AAMax training. We compare a nonlinear adapter with GELU, a nonlinear adapter with ReLU, and a two-layer linear adapter. Results show that adapter choice has little impact on performance, with all variants yielding similar alignment and reconstruction quality.

| | Low-Level | | | | High-Level | | | |
|---|---|---|---|---|---|---|---|---|
| Method | PixCorr ↑ | SSIM ↑ | Alex(2) ↑ | Alex(5) ↑ | Incep ↑ | CLIP ↑ | Eff ↓ | SwAV ↓ |
| Subject 2 FT 1HR AAMAX linear | 0.110 | 0.277 | 79.70% | 88.61% | 80.35% | 79.88% | 0.815 | 0.486 |
| Subject 2 FT 1HR AAMAX GELU | 0.112 | 0.259 | 78.75% | 88.04% | 78.99% | 78.28% | 0.824 | 0.513 |
| Subject 2 FT 1HR AAMAX ReLU | 0.101 | 0.273 | 80.44% | 88.18% | 78.72% | 78.17% | 0.827 | 0.508 |
| Subject 2 FT 1HR AAMAX 2 linear | 0.107 | 0.268 | 78.28% | 87.12% | 78.78% | 78.62% | 0.827 | 0.504 |

## D  GENERALIZING TO DIFFERENT REFERENCE SUBJECTS

We conducted additional experiments (Table 7) to examine how the choice of reference subject influences fine-tuning performance. In these experiments, Subjects 2, 5, and 7 were used as reference

subjects, and Subject 1 was fine-tuned with one hour of data. The results show that the reference subject does affect reconstruction quality: Subject 5, which typically achieves the strongest individual reconstructions, yields the best fine-tuning performance, while Subject 7 produces the weakest, consistent with its lower single-subject performance. Nonetheless, Adapter Alignment remains effective in all cases, as AAMax consistently outperforms baseline fine-tuning.

Table 7: Effect of reference subject choice on limited-data fine-tuning. We compare AAMax and normal fine-tuning (160 epochs) when Subjects 2, 5, and 7 are used as reference subjects and Subject 1 is fine-tuned. AAMax consistently outperforms normal fine-tuning regardless of the reference, though overall performance reflects the quality of the chosen reference subject.

| | Low-Level | | | | High-Level | | | |
|---|---|---|---|---|---|---|---|---|
| Method | PixCorr ↑ | SSIM ↑ | Alex(2) ↑ | Alex(5) ↑ | Incep ↑ | CLIP ↑ | Eff ↓ | SwAV ↓ |
| Subject 1 FT From Subject 2 1HR Baseline | 0.007 | 0.268 | 75.55% | 84.00% | 75.20% | 74.60% | 0.862 | 0.520 |
| Subject 1 FT From Subject 2 1HR AAMAX | **0.06** | **0.277** | **81.13%** | **88.99%** | **79.71%** | **78.89%** | **0.824** | **0.494** |
| Subject 1 FT From Subject 5 1HR Baseline | 0.094 | 0.239 | 71.54% | 80.10% | 73.11% | 74.02% | 0.875 | 0.537 |
| Subject 1 FT From Subject 5 1HR AAMAX | **0.105** | **0.259** | **76.77%** | **86.20%** | **80.08%** | **80.14%** | **0.828** | **0.496** |
| Subject 1 FT From Subject 7 1HR Baseline | 0.010 | 0.266 | 74.96% | 81.55% | 71.71% | 73.72% | 0.874 | 0.528 |
| Subject 1 FT From Subject 7 1HR AAMAX | **0.012** | **0.283** | **79.58%** | **87.78%** | **77.48%** | **77.87%** | **0.835** | **0.502** |

# E  IMAGE SELECTION ALGORITHM ABLATIONS

## E.1  ALGORITHM COVERAGE

We next evaluate how well the image selection algorithm covers the common space compared to random sampling. Using the top 20 eigenvectors, we discretize each dimension into bins and, for a subset of $K = 250$ images, count the number of empty bins (bins without a selected image). Random selection leaves, on average, around 200 empty bins with a standard deviation of about 10 across 1,000 trials. In contrast, our algorithm consistently reduces this number to 15 empty bins. A permutation test yields a p-value of 0.0002, confirming that the improved coverage is statistically significant. This demonstrates that our algorithm selects images that span the representation space more uniformly along high-variance directions. It is important to note that the number of empty bins in our algorithm remain relatively similar when changing the number of top eigenvectors.

## E.2  VARYING THE EIGENVECTOR DIMENSION

We conducted an ablation study on the image selection algorithm to evaluate the effect of varying the number of principal eigenvector dimensions used in the binning process. Specifically, we compared subsets selected using the top 10, 20, 30, and 40 eigenvectors. As shown in Table 8, performance is sensitive to this choice: 20 eigenvectors yield the strongest results, with a secondary improvement at 40, while 10 and 30 dimensions perform relatively worse. Based on these findings, we adopt 20 eigenvectors for all main experiments in the paper.

# F  COMPARING DIFFERENT ALIGNMENT TRAINING STRATEGIES

To directly compare our approach against related methods, we conducted an experiment fine-tuning Subject 2 with 250 training images under three different training regimes (Table 9). First, we evaluated AA Step 1 only, which corresponds to the setting of Ferrante et al. (Ferrante et al., 2024), where adapters are overfitted to the common space without any subsequent end-to-end training. Second, we implemented a variant inspired by MindBridge (Wang et al., 2024), where the adapter remains unfrozen while the MLP and prior are frozen, and the full end-to-end training is carried out using a combined MSE, CLIP, and prior loss. Finally, we compared these baselines to our full AAMax training regime, which explicitly aligns adapters in the first phase and then fine-tunes the entire pipeline end-to-end.

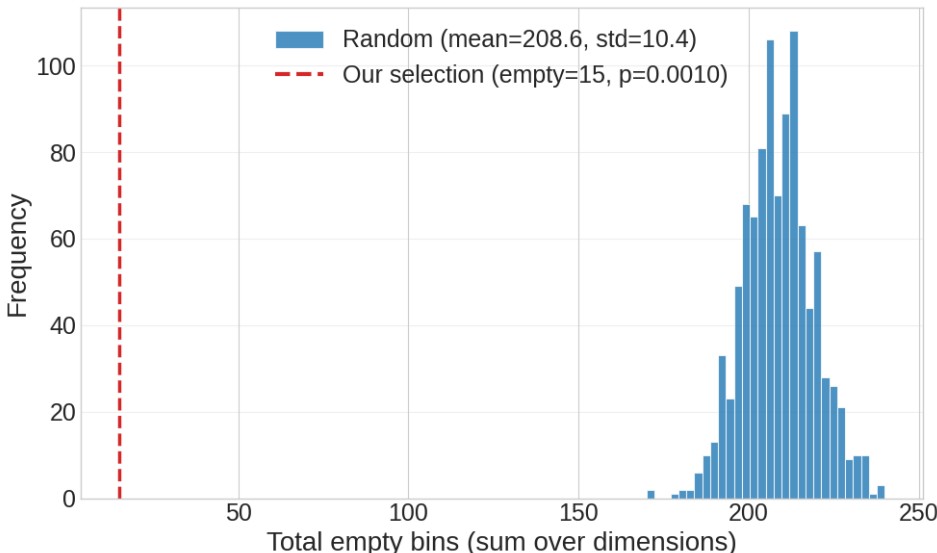

Figure 6: **Coverage of the image selection algorithm compared to random sampling.** We project embeddings onto the top 20 eigenvectors and count empty bins across 250 selected images. The histogram shows empty-bin counts for 1,000 random subsets, while the dashed line marks our algorithm's selection. Our method leaves only 15 bins empty, whereas random sampling leaves $\approx 200 \pm 10$ on average, indicating significantly more uniform coverage of the representation space ($p = 0.0002$).

Table 8: Ablation on the number of eigenvector dimensions used for image selection. We compare random image selection and Adapter Alignment with images selected using our algorithm across different eigenvector settings. Results are reported for Subject 1 as the reference and Subject 2 fine-tuned with 250 images under AAMax training, averaged over 3 runs. The best performance is obtained with 20 eigenvectors, which is used in all main experiments.

| Method | Low-Level | | | | High-Level | | | |
|---|---|---|---|---|---|---|---|---|
| | PixCorr ↑ | SSIM ↑ | Alex(2) ↑ | Alex(5) ↑ | Incep ↑ | CLIP ↑ | Eff ↓ | SwAV ↓ |
| Random Selection | 0.112 | 0.259 | 78.75% | 88.04% | 78.99% | 78.28% | 0.824 | 0.513 |
| 10 EV Dimensions | 0.114 | 0.303 | 78.99% | 86.53% | 77.30% | 75.23% | 0.850 | 0.541 |
| 20 EV Dimensions | 0.102 | 0.267 | 81.41% | 89.55% | 80.43% | 80.33% | 0.823 | 0.496 |
| 30 EV Dimensions | 0.112 | 0.275 | 79.60% | 88.03% | 78.58% | 76.89% | 0.850 | 0.537 |
| 40 EV Dimensions | 0.118 | 0.285 | 81.78% | 89.61% | 81.49% | 79.55% | 0.810 | 0.491 |

The results show that AAMax achieves the strongest performance across both low-level and high-level metrics, outperforming both AA Step 1 and the MindBridge-style end-to-end setup. Importantly, this experiment was run under identical architectural settings and without auxiliary enhancements such as additional synthetic data or architectural add-ons. This isolates the effect of the training paradigm itself, demonstrating that when compared purely on training objectives, AAMax provides the most effective strategy for fine-tuning new subjects.

## G    EXTENDED LIMITED DATA EXPERIMENTS

To further validate the efficiency of Adapter Alignment, we repeated our limited-data experiments using Subject 5 as the reference and Subject 1 as the fine-tuning subject. We evaluate fine-tuning with 1, 2, and 4 hours of data and compare baseline end-to-end training to AAMax. As always, AAMax first performs Step 1 adapter alignment by overfitting the adapter, followed by end-to-end fine-tuning. We track performance after 1 epoch of fine-tuning and after 160 epochs of fine-tuning. Even after a single epoch, AAMax consistently outperforms baseline models trained for 160–200 epochs.

Table 9: Comparison of different training regimes for fine-tuning Subject 2 with 250 training images. We evaluate (1) AAMax (full Adapter Alignment with staged alignment + end-to-end fine-tuning), (2) AA Step 1 only (adapter overfitting without subsequent end-to-end training), and (3) End-to-End with MSE+Prior+CLIP Loss (MLP Frozen) (reset-tuning). AAMax achieves the best performance across both low-level and high-level metrics, demonstrating that explicit staged alignment followed by end-to-end fine-tuning is more effective than either alignment-only or combined end-to-end training when compared under identical architectural conditions.

| Method | Low-Level | | | | High-Level | | | |
| --- | --- | --- | --- | --- | --- | --- | --- | --- |
| | PixCorr ↑ | SSIM ↑ | Alex(2) ↑ | Alex(5) ↑ | Incep ↑ | CLIP ↑ | Eff ↓ | SwAV ↓ |
| AAMAX | 0.112 | 0.259 | **78.75%** | **88.04%** | **78.99%** | **78.28%** | **0.824** | 0.513 |
| AA Step 1 only | 0.008 | 0.240 | 77.14% | 87.04% | 77.00% | 75.43% | 0.857 | 0.510 |
| End-to-End with MSE+Prior+CLIP Loss (MLP Frozen) | 0.007 | 0.238 | 76.40% | 86.42% | 77.72% | 75.04% | 0.852 | **0.507** |

We also track performance over fine-tuning epochs for all three data conditions (1, 2, and 4 hours). As shown in Figure 7, Adapter Alignment converges rapidly and maintains a consistent advantage throughout training, while baseline training requires hundreds of epochs to reach comparable accuracy. These results reinforce that AAMax substantially reduces both the data and computational requirements for fine-tuning new subjects.

Table 10: Quantitative results on fine-tuning a new subject with limited data. FT=Fine tuning. The first section shows the result of fine-tuning with 250 (1 hour) images after 1 epoch and after 160 epochs on the baseline and using AAMax. The second section compares performance on 500 (2 hours) images and the third section compares performance on 1000 (4 hours) images. In all cases AAMax, significantly outperforms the baseline. The greyed out low level metrics are not optimized for in these experiments.

| Method | Low-Level | | | | High-Level | | | |
| --- | --- | --- | --- | --- | --- | --- | --- | --- |
| | PixCorr ↑ | SSIM ↑ | Alex(2) ↑ | Alex(5) ↑ | Incep ↑ | CLIP ↑ | Eff ↓ | SwAV ↓ |
| Subject 1 FT 1HR Baseline E1 | 0.031 | 0.234 | 58.31% | 62.56% | 58.51% | 58.71% | 0.947 | 0.611 |
| Subject 1 FT 1HR AAMAX E1 | 0.095 | 0.254 | **77.20%** | 85.98% | 78.80% | 79.39% | 0.833 | 0.503 |
| Subject 1 FT 1HR Baseline E160 | 0.094 | 0.239 | 71.54% | 80.10% | 73.11% | 74.02% | 0.875 | 0.537 |
| Subject 1 FT 1HR AAMAX E160 | **0.105** | 0.259 | 76.77% | **86.20%** | **80.08%** | **80.14%** | **0.828** | **0.496** |
| Subject 1 FT 2HR Baseline E1 | 0.049 | 0.242 | 62.41% | 68.06% | 62.64% | 64.80% | 0.925 | 0.583 |
| Subject 1 FT 2HR AAMAX E1 | 0.112 | 0.275 | 79.46% | 88.29% | **83.28%** | 82.61% | 0.803 | 0.485 |
| Subject 1 FT 2HR Baseline E160 | 0.110 | 0.270 | 75.32% | 84.62% | 76.66% | 78.55% | 0.845 | 0.516 |
| Subject 1 FT 1HR AAMAX E160 | **0.114** | **0.285** | **80.00%** | **88.50%** | 83.19% | **83.05%** | **0.799** | **0.480** |
| Subject 1 FT 4HR Baseline E1 | 0.079 | 0.241 | 65.84% | 73.04% | 64.19% | 68.10% | 0.914 | 0.574 |
| Subject 1 FT 4HR AAMAX E1 | **0.129** | 0.265 | 80.83% | 90.93% | **86.66%** | 86.17% | 0.767 | 0.460 |
| Subject 1 FT 4HR Baseline E160 | 0.124 | 0.253 | 77.86% | 89.18% | 83.10% | 83.26% | 0.801 | 0.480 |
| Subject 1 FT 4HR AAMAX E160 | 0.127 | **0.273** | **81.73%** | **91.29%** | 86.23% | **86.26%** | **0.763** | **0.453** |

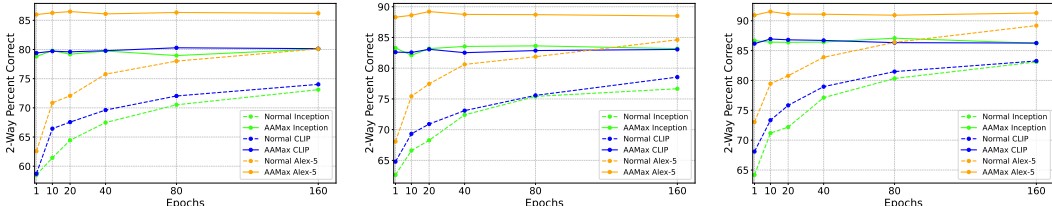

Figure 7: Comparing finetuning reconstruction performance over epochs with limited data. From left to right: (a) Finetuning Subject 1 with 250 images or 1 hour of data (b) Finetuning Subject 1 with 500 images or 2 hours of data (c) Finetuning Subject 1 with 1000 images or 3 hours of data. AAMax not only outperforms normal end-to-end training, but the performance at Epoch 1 is better than the baseline's performance at Epoch 160.

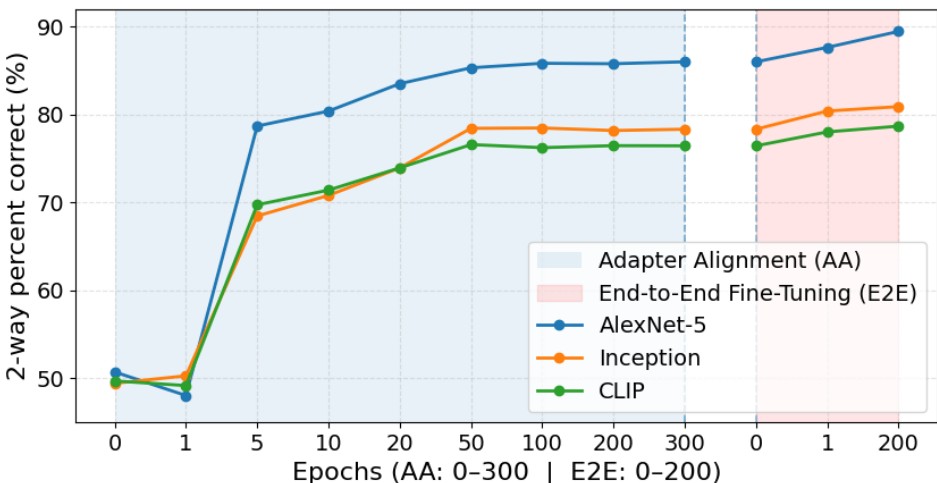

Figure 8: Reconstruction performance across epochs for Subject 2 fine-tuned with 1 hour of data using Subject 1 as the reference. The shaded regions denote Step 1 (adapter alignment) and Step 2 (end-to-end fine-tuning). Performance steadily increases during adapter alignment but plateaus short of optimal results. Transitioning to end-to-end fine-tuning yields an immediate jump in accuracy and ultimately outperforms adapter-only training. The dashed lines indicate the transition from AA to E2E. Performance at Epoch 300 of AA is the same as Epoch 0 of E2E, since no further fine-tuning has been done yet.

## H ABLATION STUDY ON THE TWO PHASES OF ADAPTER ALIGNMENT FINE-TUNING

We analyze the role of Step 1 (adapter alignment) relative to full end-to-end fine-tuning. In this experiment, we fine-tuned Subject 2 with 1 hour of data using Subject 1 as the reference. Figure 8 illustrates the reconstruction performance across epochs for both phases.

During Step 1, where only the adapter is optimized, performance improves steadily as the adapter overfits to the shared images. This phase alone provides a substantial boost over the unaligned baseline, and after sufficient epochs the adapter can achieve strong alignment across subjects. However, the improvement plateaus, as Step 1 only modifies the adapter and does not allow deeper components of the pipeline to adapt.

Introducing end-to-end fine-tuning after adapter alignment leads to an immediate jump in performance. Even after a single epoch of end-to-end updates, the model surpasses what hundreds of adapter-only epochs can achieve. We stop at around 200 epochs as the relative gain plateaus again. While the improvement in performance by a few percent from end-to-end fine-tuning might not seem significant when compared to the gain from Step 1, it can make the difference between coherent images and gibberish when working with limited data and when reconstruction fidelity is low.

## I ADDITIONAL SCALING ANALYSIS

The architectural setup used by MindEye2 leverages a 4096-dimensional shared space with BigCLIP and Stable Diffusion XL (SDXL), requiring substantially greater computational resources than our main experiments, which employ Versatile Diffusion. To evaluate whether the benefits of Adapter Alignment (AAMax) persist across different model capacities, we conducted a scaling law analysis varying both the dimensionality of the shared space and the choice of generative backbone.

### I.1 EMPIRICAL OBSERVATIONS

1. **Architectural Improvements at 1024 Dimensions:** Using a 1024-dimensional shared space, we find that BigCLIP + SDXL outperforms Versatile Diffusion, confirming that

a higher-capacity CLIP space and diffusion model contribute to improved reconstruction performance.

2. **Scaling Behavior with Versatile Diffusion:** We vary the shared space dimensionality from $512 \rightarrow 1024 \rightarrow 4096$, using adapter alignment (AAMax) fine-tuning and observe consistent improvements in reconstruction quality as dimensionality increases.

These findings indicate that AAMax is robust to architectural choice and continues to provide benefits regardless of the CLIP encoder or diffusion model employed.

## I.2    Projection to 4096 Dimensions with Stable Diffusion XL

The key insight from these experiments is that **adapter alignment is a training paradigm, not an architectural modification**. Our results demonstrate that AA is agnostic to the representational scale and generative backbone, and scales effectively with larger models. While large-scale pipelines like MindEye2 set upper bounds on reconstruction quality under high compute budgets, adapter alignment provides a complementary framework that improves efficiency and accessibility. Importantly, its benefits persist even when extended to high-capacity architectures, making it a broadly applicable strategy for fMRI-based reconstruction.

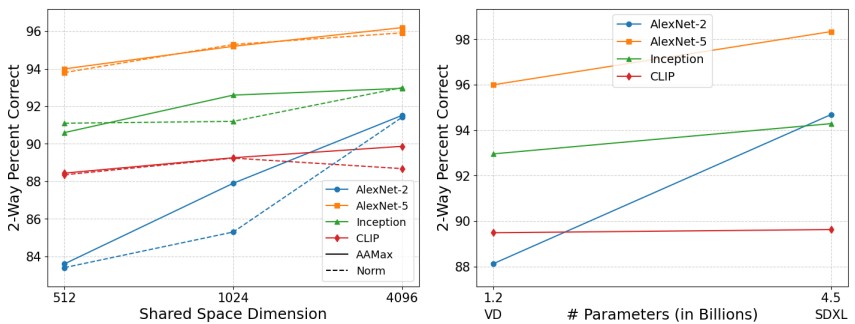

Figure 9: Comparing performance with scale for AAMax. **Left:** AAMax vs regular finetuning performance over increasing shared space size. AAMax scales with shared space dimensionality. **Right:** AAMax finetuning performance at 1024 shared space with Versatile Diffusion and Stable Diffusion XL. All metrics improve when moving to a larger diffusion model.

## J    Common Space Empirical Analysis

We run several empirical tests to validate the effectiveness of Adapter Alignment in aligning the common space across subjects. For any test that requires a one-to-one mapping we use the common images else we use the test set to perform our analysis.

## J.1    Eigenvector Analysis

To first investigate the structural alignment of embeddings across subjects, we analyzed the subject-wise cosine similarity of the principal eigenvectors in the embedding spaces produced by two techniques: the standard end-to-end training (Baseline) and our proposed approach (AAMax). For both techniques, we extracted the top five principal eigenvectors from the test data for each subject and computed the cosine similarity between the eigenvectors from different subjects. In normal end-to-end training, we observed consistently low cosine similarity between all but the first principal eigenvector of different subjects, suggesting that the primary directions of variance captured by the embeddings were highly subject-specific and lacked alignment across subjects. This indicates that the embedding space learned in this approach does not generalize well to a common structure, resulting in subject-specific representations that diverge significantly in terms of principal variance.

In contrast, AAMax training showed significantly higher cosine similarity between the principal eigenvectors of different subjects. This indicates that AAMax effectively aligns the embeddings

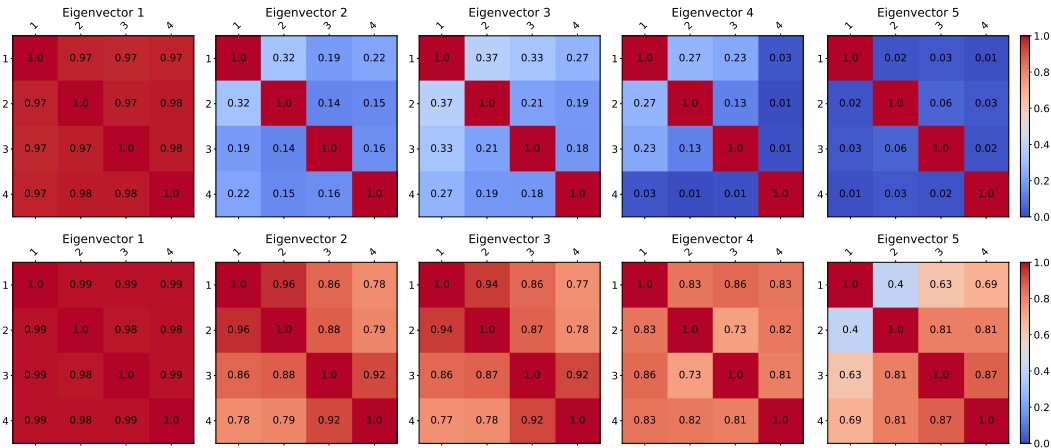

Figure 10: Comparing eigenvector cosine similarity for the test data across subjects in both training techniques. We use the first 5 principal eigenvectors. Row 1 shows the results for normal end-to-end training. Except for the first eigenvector, cosine similarity is very low across subjects. Row 2 shows the results for AAMax training. Cosine similarity is high even at the 5th principal eigenvector.

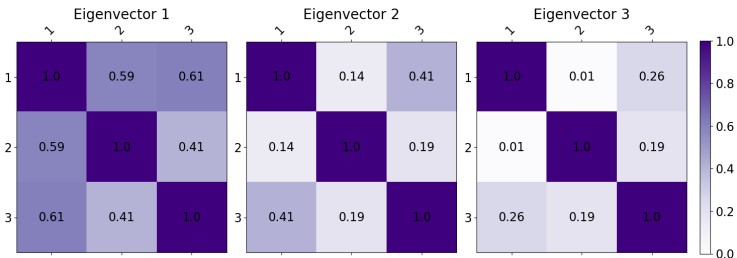

Figure 11: Comparing eigenvector cosine similarity for the test data across datasets in both training techniques. 1. Reference Subject 1 NSD 2. Normal Finetune Subject 1 THINGS 3. AAMax Finetune Subject 1 THINGS. AAMax training generates better aligned embedding spaces.

across subjects in terms of the primary directions of variance, achieving a shared representation space that better captures common structural features. The high similarity across subjects reflects AAMax's ability to encode the principal semantic and structural patterns consistently, making it more robust and generalizable compared to the normal end-to-end method.

We also perform a similar analysis across datasets in Figure 11. We compare the alignment between Subject 1 from NSD and Subject 1 from THINGS when the latter has been finetuned on a model that was pre-trained with the former. We present the difference between both training techniques. AAMax training generates better aligned embedding spaces.

## J.2 MEAN SQUARED ERROR

We compare the end-to-end training approach with AAMax by evaluating the MSE between subjects using shared images after training as shown in Figure 12. In the end-to-end model, we observe relatively high MSE between subjects, indicating poor alignment in the common space. In contrast, the MSE for AAMax is an order of magnitude lower, demonstrating a much stronger alignment of subject-specific embeddings. This result suggests that AAMax significantly enhances alignment across subjects, making it far more effective than the normal approach in bringing embeddings closer together in the common space.

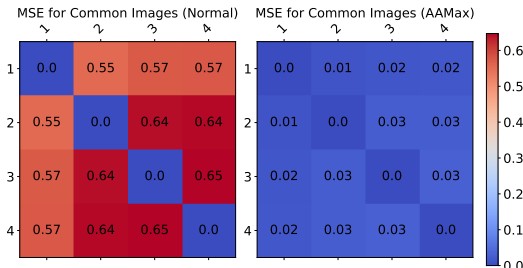

Figure 12: Comparing Mean Squared Error for the shared images across subjects in both training techniques. The MSE is compared after training is completed in both techniques. MSE for AAMax is an order of magnitude lower than the baseline

### J.3 k-Nearest Neighbor Consistency

We also evaluate the k-nearest neighbors (k-NN) consistency (Figure 13) between subjects for shared images after training, using 50 nearest neighbors as the metric. In the end-to-end model, the k-NN consistency is relatively low, indicating that the nearest neighbors of embeddings from one subject do not align well with those from another subject. However, with AAMax, the k-NN consistency significantly improves, demonstrating much better alignment of subject-specific embeddings. AAMax consistently outperforms the normal approach, highlighting its effectiveness in preserving semantic relationships across subjects in the common space.

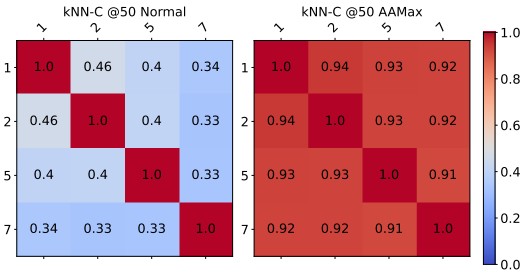

Figure 13: Comparing kNN consistency @ 50 for the shared images across subjects in both training techniques. The kNN-C is compared after training is completed in both techniques. AAMax significantly outperforms the baseline.

## K Emergence of Common Concepts: Additional Details

The existence of a semantic common representation space is suggested from the emergence of semantic categories in this space. For this, we extracted the most significant left singular vectors from the ridge regression weight matrix. This matrix was extracted from the pre-trained MindEye2 (Scotti et al., 2024) model which was trained using the 1024-dimensional common space. Subsequently, we projected the 1024-dimensional embeddings of each image onto these vectors and sampled from the two extremes (set 1 and set 3) and the middle (set 2). In order to obtain an objective classification, we presented these sets of images to GPT-4o (Achiam et al., 2023) and asked it to find the semantic categories that are well represented in set 1, somewhat represented in set 2 and not represented in set 3. Results for the first five singular vectors are shown in Table 11. As is evident, images in the principal dimensions of the representation space have a common theme.

Previous studies, such as those by Huth et al. (2012), have indicated that animal categories, including humans, are represented distinctly from non-animal categories within the brain. In our study, we observed that the common representational space also preserves these semantic properties. As demonstrated in Figure 14, after projecting the fMRI signals of Subject 2 into the common space and onto the first two singular vectors, we can see the animal vs. non-animal distinction, consistent with previous hypotheses. Extending this analysis across additional dimensions, we observed similar

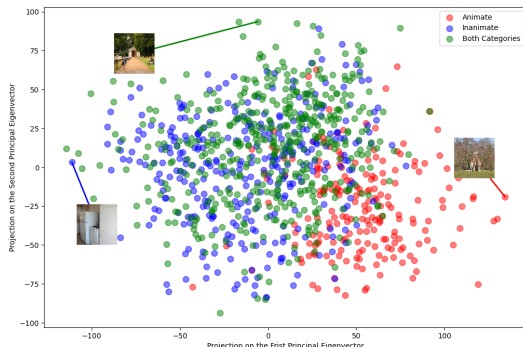

Figure 14: Visualization of the Animate/Inanimate semantic category on the two principal singular vectors. Each point (embedding) is colored based on whether its corresponding image contains 'objects' that fall into the Animate category (red), Inanimate category (blue) or both categories (green).

separations between other hypothesized semantic categories of "Social interaction," "man-made vs nature," and "biological vs non-biological."

Table 11: Concept analysis of images along principal dimensions.

| DIMENSION 1 | ANIMALS (GIRAFFES, ZEBRAS, AND ELEPHANTS), NATURE, AND OUTDOOR SCENES |
|---|---|
| DIMENSION 2 | CATS AND FOOD |
| DIMENSION 3 | KITE FLYING, AIRPLANES, AND FLYING OBJECTS |
| DIMENSION 4 | SURFING, WATER SPORTS, BEACH, AND OCEAN SCENES |
| DIMENSION 5 | AIRPLANES, FLYING OBJECTS, OUTDOOR ACTIVITIES AND EVENTS, TRANSPORTATION SCENES |

To determine if dominant semantic categories were present within these sets, we employed ChatGPT with the query: "Is there a dominant semantic category to which the majority of images in a set belong?" The methodology for defining semantic categories was consistent with that used in Huth et al. (2012). Table 11 illustrates our findings.

After projecting the fMRI signals of individual images from Subject 2 into the common space and onto the first singular vectors, we observe that certain concepts are strongly represented along these dimensions defined by the singular vectors, aligning with previous hypotheses (Figures 15, 16).

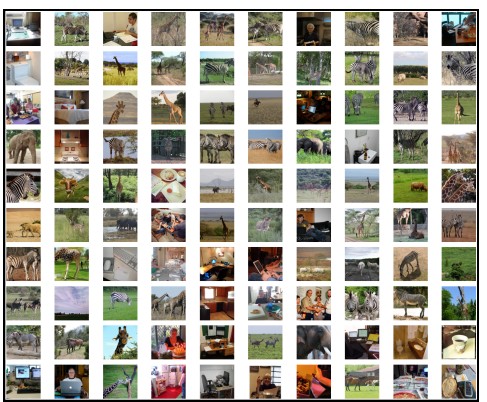 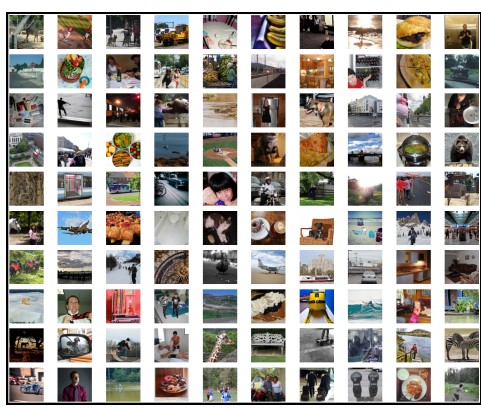

Figure 15: Comparison of images from Subject 2 with the highest and lowest projection values along the first principal left eigenvector. (a) Images whose corresponding 1024-dimensional embeddings yielded the highest projection values. (b) Images whose corresponding 1024-dimensional embeddings yielded the lowest projection values. The presence of the animate concept (e.g., giraffes, zebras, elephants) is prominent in the first set, while it fades in the second.

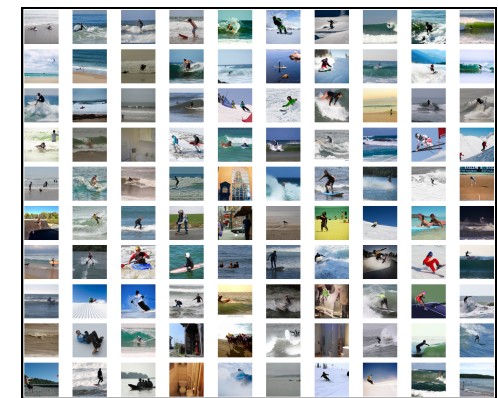 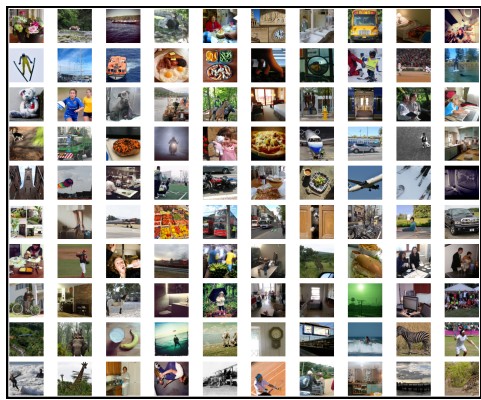

Figure 16: Comparison of images from Subject 2 with the highest and lowest projection values along the $4^{th}$ principal left eigenvector. (a) Images whose corresponding 1024-dimensional embeddings yielded the highest projection values. (b) Images whose corresponding 1024-dimensional embeddings yielded the lowest projection values. Again the presence of common concepts (e.g. surfing, water sports, beach, and ocean scenes) is prominent in the first set, while it fades in the second.

## L  NP-HARDNESS PROOF

We provide a formal proof that the bin-mapping problem is NP-hard by reduction from the Set Cover problem. The proof generalizes to an arbitrary number of bins per dimension and demonstrates the equivalence between a solution to the bin-mapping problem and a solution to the Set Cover problem.

### L.1  PROBLEM DEFINITION

The bin-mapping problem is defined as follows:

**Definition L.1** (Bin-Mapping Problem). Given a set of $n$-dimensional vectors, each mapping to one of $m$ bins per dimension, and a target $k$, the decision problem asks whether there exists a subset of $k$ vectors that spans all $m$ bins in every dimension.

### L.2  REDUCTION FROM SET COVER

We reduce the well-known Set Cover problem, which is NP-hard, to the bin-mapping problem.

**Definition L.2** (Set Cover Problem). Let $\mathcal{U}$ be a finite universe, and let $\mathcal{S} = \{S_1, S_2, \ldots, S_l\}$ be a collection of subsets of $\mathcal{U}$. The Set Cover problem asks whether there exists a subcollection $\mathcal{S}' \subseteq \mathcal{S}$ of size $k$ such that $\bigcup_{S \in \mathcal{S}'} S = \mathcal{U}$.

**Theorem L.3.** *The bin-mapping problem is NP-hard.*

*Proof.* We reduce the Set Cover problem to the bin-mapping problem as follows:

- Let $\mathcal{U} = \{1, 2, \ldots, n\}$ represent the universe of elements.

- For each subset $S_i \in \mathcal{S}$, construct an $n$-dimensional vector $V_i$ such that:

    - For every element $d \in S_i$, the vector $V_i$ places $d$ in $\text{Bin}_1$ in the corresponding dimension.
    - For all $d \notin S_i$, the vector $V_i$ places $d$ in $\text{Bin}_2$.

- Add two special vectors $N_2$ and $N_3$:

    - $N_2$ covers $\text{Bin}_2$ in all dimensions.
    - $N_3$ covers $\text{Bin}_3$ in all dimensions.

**Key Properties of the Construction:**

1. Each $V_i$ corresponds to a subset $S_i \in \mathcal{S}$.

2. Any collection of input subsets $\mathcal{S}'$ leaves a hole in $\mathrm{Bin}_2$ and $\mathrm{Bin}_3$ in at least one dimension.

3. The special vectors $N_2$ and $N_3$ are necessary to fill these holes.

4. A set of vectors $\mathcal{V}' \subseteq \mathcal{V}$ forms a bin cover if and only if the corresponding subset $\mathcal{S}'$ forms a set cover.

**Correctness:** A solution to the Set Cover problem maps directly to a solution to the Bin-Mapping problem:

- If $\mathcal{S}'$ is a set cover, then $\mathcal{S}' \cup \{N_2, N_3\}$ forms a bin cover.

- Conversely, if $\mathcal{V}'$ is a bin cover, then $\mathcal{V}' \setminus \{N_2, N_3\}$ corresponds to a set cover.

Since the Set Cover problem is NP-hard, and we have reduced it to the bin-mapping problem, the bin-mapping problem is also NP-hard. $\qquad\square$

### L.3 PROOF OF APPROXIMATION RATIO

In this section, we prove that the gap function $\mathrm{Gap}(S)$, minimized by our greedy heuristic, satisfies the submodularity property. Submodularity guarantees the diminishing returns property, which forms the basis for the $1 - \frac{1}{e}$ approximation ratio achieved by the greedy algorithm for submodular optimization (Fisher et al., 1978).

#### L.3.1 GAP FUNCTION DEFINITION

The gap function measures the total number of uncovered bins across all eigenvector dimensions:

$$\mathrm{Gap}(S) = \sum_{j=1}^{d} \mathrm{Gap}(S, j),$$

where $\mathrm{Gap}(S, j)$ is the number of bins in dimension $j$ not covered by the subset $S$. A bin in dimension $j$ is covered if any image $i \in S$ falls into that bin.

#### L.3.2 SUBMODULARITY PROPERTY

A function $f(S)$ is submodular if it satisfies the diminishing returns property (Schrijver et al., 2003):

$$f(A \cup \{x\}) - f(A) \geq f(B \cup \{x\}) - f(B), \quad \forall A \subseteq B, \; x \notin B.$$

For the gap function, this translates to:

$$\mathrm{Gap}(A \cup \{x\}) - \mathrm{Gap}(A) \geq \mathrm{Gap}(B \cup \{x\}) - \mathrm{Gap}(B),$$

where $A \subseteq B$ and $x \notin B$.

#### L.3.3 PROOF OF SUBMODULARITY

Let us consider a single dimension $j$ and define $\Delta_A(x, j)$ as the number of bins in $j$ covered by $x$ but not by $A$:

$$\Delta_A(x, j) = \mathrm{Gap}(A, j) - \mathrm{Gap}(A \cup \{x\}, j).$$

Adding an image $x$ to a subset $A$ can only cover bins that are not already covered. Since $A \subseteq B$, all bins covered by $A$ are also covered by $B$. Therefore, the additional bins covered by $x$ when added to $A$ are at least as many as those covered when adding $x$ to $B$:

$$\Delta_A(x, j) \geq \Delta_B(x, j), \quad \forall j.$$

Summing over all dimensions, the total reduction in the gap function satisfies:

$$\mathrm{Gap}(A \cup \{x\}) - \mathrm{Gap}(A) = \sum_{j=1}^{d} \Delta_A(x, j),$$

$$\text{Gap}(B \cup \{x\}) - \text{Gap}(B) = \sum_{j=1}^{d} \Delta_B(x, j).$$

Since $\Delta_A(x, j) \geq \Delta_B(x, j)$ for all $j$, we conclude:

$$\text{Gap}(A \cup \{x\}) - \text{Gap}(A) \geq \text{Gap}(B \cup \{x\}) - \text{Gap}(B).$$

Thus, $\text{Gap}(S)$ satisfies the submodularity property.

### L.3.4 MONOTONICITY OF GAP($S$)

The gap function $\text{Gap}(S)$ is also monotone non-increasing because adding more elements to $S$ can only reduce (or leave unchanged) the number of uncovered bins:

$$\text{Gap}(A \cup \{x\}) \leq \text{Gap}(A).$$

### L.3.5 CONCLUSION

Since $\text{Gap}(S)$ is both submodular and monotone, the greedy algorithm applied to minimize the gap function achieves a $1 - \frac{1}{e}$ approximation ratio (Fisher et al., 1978).

### L.4 GREEDY IMAGE SELECTION ALGORITHM

Algorithm 1 presents the greedy heuristic for bin-coverage that is utilized in the data selection method.

---

**Algorithm 1** Greedy Heuristic for Bin Coverage

---

**Require:** Set of images $\mathcal{I}$, parameter $w$
**Ensure:** $S$: Subset of images covering all bins
 1: Initialize $S \leftarrow \emptyset$
 2: Define $Gap(S, j)$: Number of empty bins in dimension $j$
 3: Define $Gap(S) \leftarrow \sum_j Gap(S, j)$
 4: **for** each dimension $j = 1, 2, \ldots, d$ **do**
 5:     Compute the number of bins $B_j = \left\lfloor w \cdot \frac{\lambda_j}{\lambda_1} \right\rfloor$
 6: **end for**
 7: **while** $Gap(S) > 0$ **do**
 8:     Select $i \in \mathcal{I} \setminus S$ minimizing $Gap(S \cup \{i\})$
 9:     Update $S \leftarrow S \cup \{i\}$
10: **end while**

---

## M TRAINING TIME

To show the efficiency of adapter alignment we computed the run time per epoch for each step of the training process. We show the AAMax takes very little extra train time to perform step 1 of the process and End-to-end fine-tuning has essentially the exact same runtime between the baseline method and AAMax. Our results are presented in Table 12.

## N HIGH-LEVEL PIPELINE TRAINING PROCESS

The high-level pipeline aims to map input fMRI signals to the CLIP embedding space. The training process differs based on whether the subject is the reference subject or a new subject being aligned to the reference space.

### N.1 TRAINING THE REFERENCE SUBJECT

For the reference subject, the loss function combines Diffusion Prior Loss and CLIP Loss. The overall loss for the reference subject is:

$$\mathcal{L}_{\text{ref}} = \lambda_1 \mathcal{L}_{\text{Prior}} + \lambda_2 \mathcal{L}_{\text{CLIP}}$$

where $\lambda_1$ and $\lambda_2$ are weighting coefficients for the respective loss components.

Table 12: Comparison of Training Times for Baseline and AAMax Training Methods. Numbers are averaged over all 3 fine tune subjects (2,5,7). The Adapters are first allowed to overfit by training them upto 400 epochs using MSE loss. This takes around a minute. Then end-to-end alignment is performed. It is important to note that besides the addition of the MSE Loss at the adapter level, both pipelines are running the same architecture and data and performance is thus very similar. The final time given is the time to train up to 160 epochs. But AAMax training can outperform 160 epochs of Baseline training with just one epoch of end-to-end alignment.

| Training Method | Adapter Alignment (Total Time) | End-to-End Fine tuning | | Total Time to Train |
|---|---|---|---|---|
| | | Time per Epoch | Total Time (160 epochs) | |
| Baseline | - | 221.4 seconds | 590.4 minutes | 590.4 minutes |
| AAMax Training | 67 seconds | 223.61 seconds | 596.21 minutes | 597.4 minutes |

### N.2    TRAINING A NEW SUBJECT

For a new subject, the training involves two stages:

**Stage 1: Adapter-Level Alignment.**    Initially, the adapter is trained using the Mean Squared Error (MSE) loss:

$$\mathcal{L}_{\text{adapter}} = \mathcal{L}_{\text{MSE}}(\mathbf{z}_{\text{adapter, new}}, \mathbf{z}_{\text{adapter, ref}})$$

where $\mathbf{z}_{\text{adapter, new}}$ and $\mathbf{z}_{\text{adapter, ref}}$ are the adapter-level embeddings for the new and reference subjects, respectively, on the shared common images.

**Stage 2: End-to-End Training.**    After aligning the adapter, the entire pipeline is trained end-to-end. During this stage:

- For non-common images, Prior Loss and CLIP Loss are applied to align the new subject's output embeddings to the CLIP space.
- For common images, the MSE Loss is applied at the adapter level to retain the alignment to the reference subject as much as possible.

The overall loss function in this stage is:

$$\mathcal{L}_{\text{new}} = \begin{cases} \lambda_1 \mathcal{L}_{\text{Prior}} + \lambda_2 \mathcal{L}_{\text{CLIP}}, & \text{for non-common images} \\ \lambda_1 \mathcal{L}_{\text{Prior}} + \lambda_2 \mathcal{L}_{\text{CLIP}} + \lambda_3 \mathcal{L}_{\text{MSE}}(\mathbf{z}_{\text{new}}, \mathbf{z}_{\text{ref}}), & \text{for common images} \end{cases}$$

where $\mathbf{z}_{\text{adapter, new}}$ and $\mathbf{z}_{\text{adapter, ref}}$ are the adapter-level embeddings for the new and reference subjects, respectively, on the shared common images.

## O    LOW-LEVEL PIPELINE TRAINING PROCESS

The low-level pipeline aims to map input fMRI signals to the latent space of an Autoencoder (AE), which is trained to compress the NSD train images into a 1024-dimensional latent representation and reconstruct them. The training process differs based on whether the subject is the reference subject or a new subject being aligned to the reference space.

### O.1    TRAINING THE REFERENCE SUBJECT

For the reference subject, the goal is to map the fMRI signals to the AE latent space using the Mean Squared Error (MSE) loss. The AE latent representations of the corresponding images serve as the ground truth.

The loss function for the reference subject is:

$$\mathcal{L}_{\text{ref}} = \mathcal{L}_{\text{MSE}}(\mathbf{z}_{\text{fMRI}}, \mathbf{z}_{\text{AE}})$$

where $\mathbf{z}_{\text{fMRI}}$ is the mapped output embedding from fMRI signals, and $\mathbf{z}_{\text{AE}}$ is the AE latent space representation of the corresponding image.

## O.2 TRAINING A NEW SUBJECT

For a new subject, the training involves two stages:

**Stage 1: Adapter-Level Alignment.** In this stage, only the adapter is trained using MSE Loss to align the new subject's adapter-level embeddings with those of the reference subject for shared common images:

$$\mathcal{L}_{\text{adapter}} = \mathcal{L}_{\text{MSE}}(\mathbf{z}_{\text{adapter, new}}, \mathbf{z}_{\text{adapter, ref}})$$

where $\mathbf{z}_{\text{adapter, new}}$ and $\mathbf{z}_{\text{adapter, ref}}$ are the adapter-level embeddings for the new and reference subjects, respectively.

**Stage 2: End-to-End Training.** In this stage, the entire pipeline is trained end-to-end. MSE Loss is applied at 2 points in the pipeline:

- At the adapter level, to maintain alignment between the new subject's embeddings and the reference subject's embeddings for common images.
- At the mapper output level, to ensure the new subject's fMRI embeddings align with the AE latent representations for corresponding images.

The loss function for this stage is:

$$\mathcal{L}_{\text{new}} = \begin{cases} \mathcal{L}_{\text{MSE}}(\mathbf{z}_{\text{fMRI, new}}, \mathbf{z}_{\text{AE}}), & \text{for non-common images} \\ \mathcal{L}_{\text{MSE}}(\mathbf{z}_{\text{fMRI, new}}, \mathbf{z}_{\text{AE}}) + \mathcal{L}_{\text{MSE}}(\mathbf{z}_{\text{adapter, new}}, \mathbf{z}_{\text{adapter, ref}}), & \text{for common images} \end{cases}$$

where:

- $\mathbf{z}_{\text{fMRI, new}}$: Output embedding for the new subject.
- $\mathbf{z}_{\text{AE}}$: AE latent space representation of the image.
- $\mathbf{z}_{\text{adapter, new}}$ and $\mathbf{z}_{\text{adapter, ref}}$: Adapter-level embeddings for the new and reference subjects, respectively.

# P  RECONSTRUCTION RESULTS

Figure 17: Reconstruction results after fine-tuning on 250 common images

# Q VISUAL COMPARISON RESULTS AMONG DIFFERENT METHODS

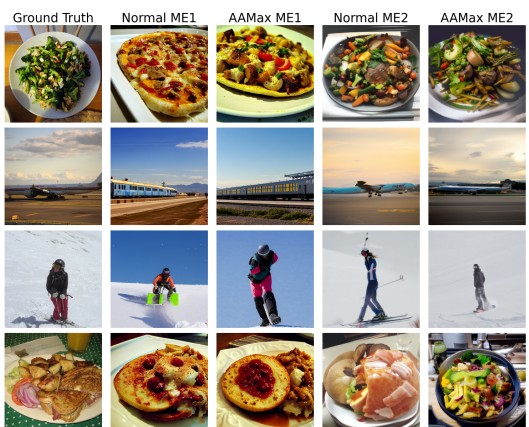

Figure 18: Visual comparison among MindEye 1 without AAMax (Normal ME1), MindEye1 with AAMax (AAMax ME1), MindEye 2 without AAMax (Normal ME2) and MindEye2 with AAMax (AAMax ME2).

# R    Transfer Experiments on Different Datasets

We conducted additional transfer experiments using the BOLD5000 Chang et al. (2019) dataset. Table 13 summarizes the results of these new experiments. They allow us to directly compare how well models trained on NSD vs. BOLD5000 serve as reference subjects for fine-tuning. All fine-tuning runs use AAMax with 500 training samples for the fine tune subject on the ME1 architecture. The reference subjects across both datasets were trained on all subject specific available data.

- Row 1 (B5K Subj1 Pre-train): Performance of a model trained from scratch on Subject 1 of BOLD5000.
- Row 2 (FT B5K Subj2 on B5K Subj1): Fine-tuning Subject 2 of BOLD5000 using Subject 1 of BOLD5000 as the reference.
- Row 3 (FT B5K Subj2 on NSD Subj1): Fine-tuning Subject 2 of BOLD5000 using NSD Subject 1 as the reference.
- Row 4 (FT NSD Subj2 on NSD Subj1): Fine-tuning NSD Subject 2 using NSD Subject 1 as the reference (the standard NSD→NSD setting).
- Row 5 (FT NSD Subj2 on B5K Subj1): Fine-tuning NSD Subject 2 using BOLD5000 Subject 1 as the reference.

Across these conditions, we consistently observe that NSD serves as a substantially stronger reference dataset. NSD-trained reference models yield higher-quality fine-tuning outcomes on both NSD and BOLD5000 target subjects. In contrast, BOLD5000-trained reference models lead to noticeably weaker reconstruction performance even when transferring within the same dataset. These results confirm the intuition that NSD's high-resolution 7T acquisitions give rise to richer, more stable subject-specific representation spaces, which in turn make NSD the most reliable and effective choice for constructing a reference model. This finding aligns with our original rationale for using NSD as the backbone dataset for the common space.

Table 13: Cross-dataset and within-dataset transfer performance between NSD and BOLD5000.

| Method | Low-Level | | | | High-Level | | | |
|---|---|---|---|---|---|---|---|---|
| | PixCorr ↑ | SSIM ↑ | Alex(2) ↑ | Alex(5) ↑ | Incep ↑ | CLIP ↑ | Eff ↓ | SwAV ↓ |
| B5k Subj1 Pre-train | 0.041 | 0.211 | 65.71% | 76.87% | 66.24% | 74.77% | 0.612 | 0.928 |
| FT B5k Subj2 on B5k Subj1 | 0.047 | 0.247 | 60.53% | 66.23% | **60.06%** | 64.63% | **0.631** | 0.952 |
| FT B5k Subj2 on NSD Subj1 | 0.053 | 0.252 | **63.32%** | **69.13%** | 59.75% | **66.07%** | 0.634 | **0.948** |
| FT NSD Subj2 on B5k Subj1 | 0.115 | 0.289 | 79.76% | 86.73% | 79.57% | 79.88% | 0.499 | 0.825 |
| FT NSD Subj2 on NSD Subj1 | 0.114 | 0.270 | **83.70%** | **91.76%** | **83.71%** | **82.00%** | **0.469** | **0.789** |

# S    Combining Adapter Alignment and Image Selection with MindAligner

We conducted a direct comparison between MindAligner and our Adapter Alignment (AA) method using the MindEye1 architecture under the same experimental protocol used throughout the paper. As before, we pretrain the full model on Subject 1 and fine-tune on Subject 2 using 1 hour of data. We evaluate three configurations:

1. **MindAligner fine-tuning:** We applied MindAligner exactly as described in (Dai et al., 2025), including the generation of categorical stimulus pairs, even though the training data contain exact shared stimuli. This procedure yields significantly more training examples than the 250-image subset used in our low-data experiments, and the model is trained using the original MindAligner voxel-space loss functions.

2. **Adapter Alignment + Image Selection:** Using our proposed image-selection algorithm, we generate a 250-image subset that spans the breadth of the principal eigenspace of Subject 1's shared representation. This ensures coverage across all principal axes of variation, enabling

AA to perform exact one-to-one alignment in the shared space. This configuration mirrors the setup used in all other AA experiments.

3. **MindAligner + Adapter Alignment + Image Selection:** MindAligner and AA operate at different levels of the pipeline. MindAligner performs coarse categorical voxel-space alignment, and AA performs fine-grained common-space alignment. This lets us build and evaluate a multi-stage pipeline. Training a new subject is decomposed into four steps:

   (a) fine-tune the BFA (Brain Feature Adapter) to reconstruct category-level fMRI for Subject 1;

   (b) apply Adapter Alignment by overfitting Subject 1's adapter on the new 1-hour dataset;

   (c) perform end-to-end training of the BFA module with the remainder of the architecture frozen;

   (d) perform AA end-to-end fine-tuning with the BFA frozen.

   This sequence allows MindAligner to establish coarse alignment at the voxel level, while AA and Image Selection refine the geometry of the shared space.

Table 14 summarizes the results. We observe that AA + Image Selection matches MindAligner. Combining MindAligner with AA and Image Selection yields the best overall performance, establishing a new benchmark for the MindEye1 architecture under this low-data setting. These results reinforce our claim that MindAligner and AA are complementary, not competing approaches: MindAligner provides categorical coarse alignment in voxel space, while AA offers precise one-to-one alignment in shared space, and Image Selection ensures that the selected subset spans the relevant eigenspace for stable mapping. Together, these stages produce the strongest cross-subject decoding performance.

Table 14: **Comparison and combination of MindAligner and Adapter Alignment.** All methods use MindEye1 with pretraining on Subject 1 and fine-tuning on Subject 2 for 1 hour of data. MindAligner is applied exactly as in the original paper using categorical stimulus pairs. AdapterAlignment (AA) uses our 250-image Image Selection subset that spans the breadth of the principal eigenspace. The combined method applies MindAligner for coarse voxel-space alignment followed by AA + Image Selection for fine-grained common-space alignment. The combined approach achieves the strongest overall performance.

| Method | Low-Level | | | | High-Level | | | |
|---|---|---|---|---|---|---|---|---|
| | PixCorr ↑ | SSIM ↑ | Alex(2) ↑ | Alex(5) ↑ | Incep ↑ | CLIP ↑ | Eff ↓ | SwAV ↓ |
| MindAligner | 0.116 | 0.280 | 82.11% | 88.68% | 81.36% | 79.69% | 0.809 | 0.483 |
| AdapterAlignment + Image Selection | 0.118 | 0.285 | 81.78% | 89.61% | 81.45% | 79.55% | 0.810 | 0.491 |
| MindAligner + AdapterAlignment + Image Selection | 0.124 | 0.284 | **82.24%** | **89.95%** | **81.49%** | **80.99%** | **0.795** | **0.480** |