# OpenReview forum: "Efficient Multi Subject Visual Reconstruction from fMRI Using Aligned Representations"
_ICLR.cc/2026/Conference — Submitted to ICLR 2026_

### Official Review · Reviewer_hpp1 · 2025-10-30

**Soundness:** 3
**Presentation:** 3
**Contribution:** 2
**Rating:** 4
**Confidence:** 4

**Summary:**

This paper introduces a new fMRI decoding method to improve the cross-subject decoding performance. Specifically, the authors employs subject-specific adapters to project new subjects into a shared reference space, then performs end-to-end fine-tuning of the remaining decoding network.  A sample selection mechanism is also leveraged to select representative samples during decoding. Experiments verify the effectiveness of the proposed method over two baselines.

**Strengths:**

- This paper is clearly written and well-organized.
- This paper proposes an adapter alignment training strategy to align new subjects to the reference subject in the low-data regime. The experiments demonstrate the effectiveness of the proposed method.

**Weaknesses:**

- Several relevant works are missing, such as MindAligner [1], which is highly related as it also aims to align new subjects to a reference space. To strengthen the paper, the authors should include a discussion that explicitly contrasts their method with these approaches. Highlighting the differences would more effectively establish the novelty and unique contribution of this research.
- While the proposed subject-specific adapters are a promising alignment solution, the subsequent end-to-end fine-tuning raises concerns regarding efficiency and practicality. If the adapter achieves effective alignment, fine-tuning may be redundant. In comparison, other methods (MindAligner [1], Ferrante et al. 2024 [2] & MindBridge) do not need additional fine-tuning. Furthermore, this fine-tuning step introduces substantial computational cost and could potentially perturb the carefully learned adapter alignment.
- The authors only compare two baselines (MindEye and MindEye2) in the main experiment. I’d like to see more baselines to justify the effectiveness of the method.
- The alignment technique in this paper requires the same stimuli for different subjects, which may not be a common requirement in practice.

[1] MindAligner: Explicit Brain Functional Alignment for Cross-Subject Visual Decoding from Limited fMRI Data

[2] Through their eyes: multi-subject brain decoding with simple alignment techniques.

**Questions:**

- Could authors provide performance comparison with more baselines?

- A key methodological concern is the modification of the data split, where common images were transferred from the test set to the training set. The comparison with previous methods is less soundable unless those baselines were also re-trained and re-evaluated on this identical, new data split. Please clarify it.

Please refer to the Weaknesses for other questions.

---

> ### Author Response · Authors · 2025-11-22
>
> We thank the reviewer for their detailed feedback. We are glad that they found our paper to be well written. We address your concerns below.
>
> ---
>
> **W1:** We thank the reviewer for pointing this out. We will add an explicit discussion in the related work contrasting MingAligner with our approach. MindAligner [1] and Adapter Alignment differ fundamentally in where and how alignment occurs:
> - MindAligner aligns subjects in voxel space using categorical priors and many-to-many mappings, driven by the absence of shared stimuli.
> - Adapter Alignment (AA) aligns in the learned common representation space after the subject-specific adapter and performs a one-to-one mapping with a two-stage alignment procedure (adapter overfitting + joint fine-tuning).
> - The methods are complementary, not competing; MindAligner performs coarse categorical alignment, while AA provides fine-grained common-space alignment.
>
> We are currently running additional experiments to compare and combine MindAligner and AAMax+Image Selection. We will update our response with the results once they are complete.
>
> ---
>
> **W2:** We thank the reviewer for raising this concern. We directly address this question through dedicated experiments in Appendix F and Appendix H of the manuscript.
> 1. **Empirical performance of two-stage alignment:** Appendix F compares AA against the alignment strategies used in MindBridge [2] and in 'Through their eyes' [3]. These experiments show that adapter-only alignment or only end-to-end finetuning is not sufficient: the two-stage AA process (adapter alignment + end-to-end fine-tuning) yields stronger better reconstruction performance than the single-stage methods used in prior work.
>
> 2. **Gains from end-to-end finetuning:** Appendix H (Fig. 8) presents performance curves across all training phases. The plots clearly show that alignment improves substantially during the AA stage, and performance improves further during end-to-end fine-tuning, rather than plateauing.
>
> 3. **Computation cost:** The end-to-end stage is far less expensive than training an entire multi-subject model from scratch. AA+E2E produces SoTA performance after a single epoch of end-to-end finetuning, and more epochs only give additional gains.
>
> ---
>
> **W3/Q1:** MindEye1 [4] and MindEye2 [5] remain the state-of-the-art pipelines for fMRI-to-image reconstruction that rely solely on architectural improvements without explicit subject alignment. Most other recent works use variants of the same two-path architecture (low-level autoencoder + high-level CLIP mapper) with small modifications to the MLP or diffusion prior. For this reason, MindEye1/2 represent the strongest and most relevant architectural baselines for evaluating training-paradigm improvements. To further strengthen the comparison and prove AA’s architecture agnosticity:
> - We are currently running experiments integrating MindAligner into our pipeline. Early results indicate that AA + Image Selection is comparable to MindAligner, and can improve performance when combined.
> - Appendix C shows that AA is robust across different adapter architectures (nonlinear, linear, and deeper adapter variants), demonstrating the method is not tied to the MindEye design.
> - Appendix I presents scaling analyses, showing that AA continues to provide improvements when varying the dimensionality of the shared space or using more powerful diffusion models, further supporting its generality beyond a specific backbone.

---

> ### Author Response · Authors · 2025-11-22
>
> **W4:** While our alignment method can use identical shared stimuli when they are available, it does not require them in practice. Our THINGS transfer experiments (Section 5) explicitly demonstrate that Adapter Alignment works even when subjects view different but semantically similar images. In this setting, the method aligns subjects effectively without relying on exact stimulus matches.
>
> The ability to align across semantically matched but non-identical stimuli is actually a key practical strength of our approach as pointed out by reviewer HkQB and we will emphasize this more clearly in the revised manuscript. Identical stimuli simply provide an upper-bound scenario, useful for establishing the best-case fine-tuning procedure and data-collection strategy for new subjects. But in typical neuroscience settings where stimuli differ across participants, AA remains fully applicable as long as a small subset of images share semantic similarity.
>
> We will expand the discussion in the revision to highlight that the method is not dependent on exact shared images and show how the THINGS results validate this in a real cross-dataset setting.
>
> ---
>
> **Q2:** We thank the reviewer for raising this point. To ensure a fully fair comparison, every baseline was re-trained and re-evaluated on the identical modified train/test split. No method uses the original NSD split.
>
> Thus, all results in the paper are computed under the same data partition, architecture settings, and training schedule, ensuring consistency across methods despite the modified split. We will clarify this explicitly in the revised manuscript.
>
> ---
> We hope we have addressed all your concerns. Please let us know if you have any additional questions.
>
> ---
>
> ### References
>
> 1.  Dai, Yuqin, et al. "MindAligner: Explicit Brain Functional Alignment for Cross-Subject Visual Decoding from Limited fMRI Data." arXiv preprint arXiv:2502.05034 (2025).
>
> 1. Wang, Shizun, et al. "Mindbridge: A cross-subject brain decoding framework." Proceedings of the IEEE/CVF Conference on Computer Vision and Pattern Recognition. 2024.
>
> 1.  Ferrante, Matteo, et al. "Through their eyes: multi-subject Brain Decoding with simple alignment techniques." Imaging Neuroscience 2 (2024): 1-21.
>
> 1. Scotti, Paul, et al. "Reconstructing the mind's eye: fmri-to-image with contrastive learning and diffusion priors." Advances in Neural Information Processing Systems 36 (2023): 24705-24728.
>
> 1. Scotti, Paul S., et al. "Mindeye2: Shared-subject models enable fmri-to-image with 1 hour of data." arXiv preprint arXiv:2403.11207 (2024).

---

### Official Review · Reviewer_HkQB · 2025-10-31

**Soundness:** 3
**Presentation:** 3
**Contribution:** 2
**Rating:** 6
**Confidence:** 3

**Summary:**

This paper presents Adapter Alignment (AA), an architecture- and paradigm-agnostic method to align the fMRI representations of different subjects to a common reference-subject space. This is applied as an efficient fine-tuning strategy of fMRI-to-image models: a model trained on one subject with maximum data is fine-tuned using AA to produce models for new / unseen subjects that perform better than previous fine-tuning baselines (e.g. MindEye2), in limited data setting. Additionally, the paper provides a greedy-based method for selecting the best subset of images to use for fine-tuning, further reducing the data requirements.

**Strengths:**

- The paper is very well written, easy to follow and understand.
- The motivation for the method and the underlying 'common subject-space hypothesis' is explicitly given and well detailed.
- The authors show that their alignment method still works even when a common set of images for all subjects is missing (i.e application to the THINGS-fMRI dataset)
- The method of analysis is clear, sound and generalises across two different types of MindEye architectures.
- The set of ablations presented is rather complete, with additional insightful visualisations.
- The performance gain in limited data setting against the presented baselines is clear and unambiguous.

**Weaknesses:**

- Although the experiments are run with different seedings and subjects, then averaged, error bars / confidence intervals are missing. Adding them would enhance the impact / strength of these results.
- There are already several studies on fine-tuning a reference model on new subjects and the present paper mostly refers only to the common-space subject-module / adapter of MindEye1/2, hence this very specific improvement related to aligning subject spaces in MindEye architectures can be seen rather of iterative nature than a stronger contribution. It could be strengthened by extending the method to more datasets rather than NSD and THINGS (e.g. Bold5K, DeepRecon), architectures other than ME1/2, and more subject-embedding techniques (concat, add, ...).
- Given the now well-know flaws of the NSD dataset (low categorical diversity, categorical leakage, ..) the use of NSD as the only reference dataset is a potentially weak point of the paper, but could be addressed by doing at least one other transfer ablation (e.g. between THINGS and Bold5K).

**Questions:**

- it is not entirely clear to me if the SVD analysis for Image selection is carried out on the 1000 fMRI representations  of the common image set (in reference subject-space), or if it is done on the CLIP embeddings of the corresponding images
- It seems that only single-subject models are trained, but it is sometimes ambiguous whether the subjects 2,5,7 are fine-tuned simultaneously with a single model (with one adapter per subject, yielding a single model for all 3 subjects), or one by one, separately (yielding 3 different models)
- Tables in the main paper gray out the pixel-wise metrics because it is said the pipeline does not optimise for these. This could be clarified so that it is clearer that the results are still valid for these metrics. It would also be interesting to report metrics for the low-level pipeline of ME1/2.
- It is not clear to me what the strategy is, in the end-to-end fine-tuning stage, for re-using common images or not. Some parts of the text mention that the common-images are used only for AA (e.g. 4.3), but it is unclear if other experiments follow this same strategy or reuse the common images in the end-to-end stage.
- There are other strategies than mapping fMRI to a common space (e.g. adding a subject-embedding at various levels in this pipeline), it would be interesting to have a discussion of these other types of strategies for embedding subject information
- The transfer results on THINGS that shows you don't need a common set of identical images across subjects, but just semantically-matching collections of images across subjects is most interesting and I believe it could be emphasised more strongly as a core contribution of the paper (i.e the method is also agnostic to having a common set of images across subjects).
- Unless I'm mistaken there is a typo at L275: it should be the 'total number of empty bins across all dimensions' for $Gap(S)$
- It is not clear to me whether the IS algorithm keeps a constant number (250) of images and tries to minimise the number of empty bins under this constraint or varies the number of images to allow for fewer empty bins. In the second case, it would be informative to have sizes of the image subset returned by the IS algorithm.

---

> ### Author Response · Authors · 2025-11-22
>
> We thank the reviewer for their detailed feedback. We are glad that they found our work to be well motivated and written. We address your concerns below. **W** refers to weakness and **Q** refers to questions.
>
> ---
>
> **W1:** We followed the standard practice in the field, where prior works such as MindEye1 [1], MindEye2 [2], MindBridge [3] and Brain Diffuser [4] do not report standard deviations. But we are in the process of reconstructing the saved checkpoints again to add the error bars. We have added error bars for subject 2 in Table 2 and will continue updating the manuscript as more runs finish.
>
> ---
>
> **W2:** We thank the reviewer for pointing this out. We would like to make a few clarifications:
> 1. While we agree that this is a rapidly growing field, there are not many works that attempt explicit alignment across subjects. Additionally our contributions include not only a novel technique for alignment but also a novel algorithm for image selection and results on cross dataset generalization. We also compare our alignment method against related techniques in Appendix F where we show that our two-stage method is the best approach.
>
> 2. Our contribution is not tied to a specific MindEye architecture. Adapter Alignment (AA) is a training paradigm that operates at the representation level, is architecture-agnostic, and provides complementary benefits regardless of the backbone. The same is true for our image-selection algorithm, which is fully model-independent and can be applied to any cross-subject decoding pipeline. While our initial experiments use the MindEye1/2 backbone to ensure comparability to prior multi-subject decoding literature, the method is not constrained by it. Most works in the literature use some form of a feedforward network to align to a CLIP space and decode with a diffusion/GAN. Our Scaling analyses in Appendix I further shows that our training paradigm scales with different architectural components and intermediate dimensions.
>
> We agree that extending to additional datasets and subject-embedding mechanisms would further highlight the generality of the approach. As noted in **W3**, NSD provides a unique advantage for initial evaluation due to its 7T resolution and high SNR, making it a strong testbed for establishing the method’s behavior under ideal conditions. Nonetheless, we fully agree that validating transfer beyond NSD is important. To address this, we are running additional experiments on BOLD5000 to assess cross-dataset and cross-scanner generalization. We will post an updated response once we have the results and update the manuscript.
>
> ---
>
> **W3:** We thank the reviewer for raising this important point. NSD offers a key advantage for evaluating reconstruction pipelines; its fMRI data are collected using a 7T scanner, providing substantially higher spatial resolution and SNR than datasets acquired at 3T (such as THINGS or BOLD5000). Because our method relies on learning fine-grained, subject-specific mappings, our working hypothesis is that NSD should yield stronger performance than lower-resolution datasets, making it a natural baseline for initial analysis. That said, we fully agree that demonstrating transfer beyond NSD would strengthen the paper. We are therefore running additional experiments using BOLD5000 as a second dataset to evaluate cross-dataset transfer and ablation performance. We will update our response once our runs finish.
>
> ---
>
> **Q1:** We thank the reviewer for giving us the opportunity to clarify. As described in Section 3.4, the SVD analysis used for image selection is performed entirely in the common space. Specifically, we decompose the adapter weight matrix of the trained reference subject, obtaining its top d singular vectors. Each of the 1000 fMRI embeddings of the common image set is then projected onto these singular vectors, producing coordinates that capture the principal axes of variability in common representation space. CLIP embeddings are not used at any stage of the image-selection procedure. The selection is based solely on the structure of the fMRI-derived common space defined by the reference subject.
>
> ---
>
> **Q2:** In all of our experiments, subjects are fine-tuned independently, not jointly. That is, for Subjects 2, 5, and 7, we perform three separate fine-tuning procedures, each producing a distinct model with its own subject-specific adapter. In the end, you would have a trained model with 2 subjects in it, a reference subject and a fine-tuned subject. The only component shared across these runs is the reference model trained on Subject 1, whose adapter embeddings define the common representation space. It is possible to add additional fine-tuning subjects to the same model as the reference adapter stays frozen during the process.

---

> ### Author Response · Authors · 2025-11-22
>
> **Q3:** We thank the reviewer for pointing this out. We will clarify in the final version that the pixel-wise metrics are grayed out only because the high-level pipeline does not explicitly optimize for them, but the reported values remain valid and comparable across methods. We will also consider re-running the experiments with explicit low-level optimization and include the corresponding ME1/ME2 low-level metrics in the revised manuscript.
>
> ---
>
> **Q4:**  In all our experiments—both with and without Adapter Alignment the common images are always included in the training process.
> - In the limited-data experiments, the only training data are the common images.
> - During end-to-end fine-tuning, common images remain part of the training set for both the normal and AAMax variants.
> - When AAMax is active, we apply an additional adapter-level MSE loss on these common images during end-to-end fine-tuning, alongside the usual diffusion-prior losses and contrastive (InfoNCE/CLIP) losses.
> - We will revise the manuscript to make this consistent across Sections 3–4 and eliminate any ambiguous phrasing.
>
> ---
>
> **Q5:** We thank the reviewer for this suggestion and agree that alternative strategies for embedding subject information are an important area to discuss. There is a growing body of work that uses subject embeddings instead of (or alongside) explicit common-space alignment. Csáky et al. [5] concatenate learned subject embeddings to group-level models, and Yin et al. [6] introduce subject-specific embeddings and subject–masker modules within multimodal brain encoders. We are surveying additional related work and will add a dedicated discussion in the Related Work section to more explicitly contrast these approaches with Adapter Alignment.
>
> On the experimental side, we conducted a related ablation by increasing the complexity of the adapter, which is conceptually similar to injecting deeper subject-specific transformations. As reported in Appendix C, moving from a single linear adapter to multi-layer adapters slightly decreases performance, and the best results are obtained with a single-layer adapter.
>
> We appreciate the reviewer for raising this discussion point and believe that richer subject-embedding strategies within the AA framework are a promising direction for future work.
>
> ---
>
> **Q6:** We thank the reviewer for highlighting this. We fully agree that the THINGS transfer experiment is an important contribution: it demonstrates that Adapter Alignment does not require identical shared images across subjects and that semantically matched stimuli are sufficient for effective alignment. We will revise the manuscript to emphasize this result more prominently as a core contribution.
>
> ---
>
> **Q7:** Thank you for pointing this out, we have fixed the typo.
>
> ---
>
> **Q8:** We thank the reviewer for the question. To clarify: in our experiments the Image Selection (IS) algorithm is required to produce a fixed subset of 250 images. For a given number of principal eigenvectors, we adjust the bin-scaling parameter $w$ so that the greedy coverage procedure yields a subset of exactly 250 images while still covering all bins across dimensions. The subset size is fixed, and $w$ is the variable controlling bin granularity to ensure full coverage.
>
> ---
>
> We hope this addresses the reviewer's concerns. We ask for their patience as we finish the runs on BOLD5K. We will update our response once we have the results. In the meantime, please let us know if you have any additional questions.

---

> ### Author Response · Authors · 2025-11-22
>
> ### References
>
> 1. Scotti, Paul, et al. "Reconstructing the mind's eye: fmri-to-image with contrastive learning and diffusion priors." Advances in Neural Information Processing Systems 36 (2023): 24705-24728.
>
> 1. Scotti, Paul S., et al. "Mindeye2: Shared-subject models enable fmri-to-image with 1 hour of data." arXiv preprint arXiv:2403.11207 (2024).
>
> 3. Wang, Shizun, et al. "Mindbridge: A cross-subject brain decoding framework." Proceedings of the IEEE/CVF Conference on Computer Vision and Pattern Recognition. 2024.
> 4. Chen, Xuhang, et al. "Brain diffuser: An end-to-end brain image to brain network pipeline." Chinese Conference on Pattern Recognition and Computer Vision (PRCV). Singapore: Springer Nature Singapore, 2023.
> 5. Csaky, Richard, et al. "Group‐level brain decoding with deep learning." Human Brain Mapping 44.17 (2023): 6105-6119.
> 6. Yin, Xuanhua, Runkai Zhao, and Weidong Cai. "Improving Multimodal Brain Encoding Model with Dynamic Subject-awareness Routing." arXiv preprint arXiv:2510.04670 (2025).

---

> ### Author Response · Authors · 2025-11-24
>
> |                         | PixCorr | SSIM  | AlexNet2 | AlexNet5 | Inception | Clip   | Swav  | EffNet |
> |---------------------------------|---------|-------|----------|----------|-----------|--------|-------|--------|
> | B5k Subj1 Pre-train             | 0.041   | 0.211 | 65.71%    | 76.87%    | 66.24%     | 74.77%  | 0.928 | 0.612  |
> | FT B5k Subj2 on B5k Subj1       | 0.047   | 0.247 | 60.53%   | 66.23%    | 60.06%     | 64.63%  | 0.952 | 0.631  |
> | FT B5k Subj2 on NSD Subj1       | 0.053   | 0.252 | 63.32%    | 69.13%    | 59.75%     | 66.07%  | 0.948 | 0.634  |
> | FT NSD Subj2 on B5K Subj1       | 0.115   | 0.289 | 79.76%    | 86.73%    | 79.57%     | 79.88%  | 0.825 | 0.499  |
> | FT NSD Subj2 on NSD Subj1       |     0.114    |    0.270   |   83.70%     |      91.76%    |      83.71%     |    82.00%    |   0.789    |   0.469     |
>
>
> We conducted additional transfer experiments using the BOLD5000 [1] dataset. Below we summarize the results of these new experiments. They allow us to directly compare how well models trained on NSD vs. BOLD5000 serve as reference subjects for fine-tuning. All fine-tuning runs use AAMax with 500 training samples for the fine tune subject on the ME1 architecture. The reference subjects across both datasets were trained on all subject specific available data.
> - Row 1 (B5K Subj1 Pre-train): Performance of a model trained from scratch on Subject 1 of BOLD5000.
>
>
> - Row 2 (FT B5K Subj2 on B5K Subj1): Fine-tuning Subject 2 of BOLD5000 using Subject 1 of BOLD5000 as the reference.
>
>
> - Row 3 (FT B5K Subj2 on NSD Subj1): Fine-tuning Subject 2 of BOLD5000 using NSD Subject 1 as the reference.
>
>
> - Row 4 (FT NSD Subj2 on NSD Subj1): Fine-tuning NSD Subject 2 using NSD Subject 1 as the reference (the standard NSD→NSD setting).
>
>
> - Row 5 (FT NSD Subj2 on B5K Subj1): Fine-tuning NSD Subject 2 using BOLD5000 Subject 1 as the reference.
>
>
> Across these conditions, we consistently observe that NSD serves as a substantially stronger reference dataset. NSD-trained reference models yield higher-quality fine-tuning outcomes on both NSD and BOLD5000 target subjects. In contrast, BOLD5000-trained reference models lead to noticeably weaker reconstruction performance even when transferring within the same dataset. These results confirm the intuition that NSD’s high-resolution 7T acquisitions give rise to richer, more stable subject-specific representation spaces, which in turn make NSD the most reliable and effective choice for constructing a reference model. This finding aligns with our original rationale for using NSD as the backbone dataset for the common space.
> We will include these BOLD5000 transfer ablations in the revised manuscript.
>
> 1. Chang, Nadine, et al. "BOLD5000, a public fMRI dataset while viewing 5000 visual images." Scientific data 6.1 (2019): 49.

---

### Official Review · Reviewer_k8hD · 2025-10-31

**Soundness:** 2
**Presentation:** 2
**Contribution:** 2
**Rating:** 2
**Confidence:** 4

**Summary:**

This paper presents an fMRI visual reconstruction method combining Adapter Alignment and a submodular greedy image selection algorithm. It aligns new subjects’ representation spaces with a pre-trained reference via lightweight adapters and selects representative training images for data efficiency. Validated on NSD and THINGS datasets, the authors claim that their method outperforms end-to-end baselines in low-data scenarios and is architecture-agnostic, compatible with different reconstruction architectures and protocols.

**Strengths:**

The paper attempts to address the low data efficiency issue in fMRI visual reconstruction, which is a practically significant problem in the field. The combination of adapter alignment and submodular greedy image selection shows some engineering ingenuity in optimizing data utilization.

**Weaknesses:**

1. Unreasonable and Unfair Dataset Split: The authors modified the standard training-test split of the NSD dataset, moving shared images from the test set to the training set to achieve a one-to-one image mapping across subjects. This operation lacks a valid justification and is unfair. In fact, semantic alignment can be fully achieved through similar images rather than identical ones.
2. Inconsistent Baseline Results Undermine Fair Comparison: The reported MindEye2 baseline results in this paper are significantly inconsistent with those in the original MindEye2 paper, with much lower values. This renders the performance comparison between the proposed method and the baseline unfair, failing to truly reflect the method’s actual effectiveness.
3. Lack of Novelty in Core Idea and Insufficient Comparison with Prior Work: The core idea of mapping new subjects to a pre-trained reference subject has already been presented in existing public research (e.g., MindAligner[1]). The authors did not conduct a systematic comparison between their method and MindAligner, nor did they clearly demonstrate their superiority in this task.
4. Unclear Research Significance and Limited Biological Insights: The research significance of the proposed method could be further clarified. In terms of decoding performance, it does not exceed the results reported in MindEye2 and MindAligner. From a biological perspective, the work does not elaborate on new insights into brain function or neural representation mechanisms, and the method leans more toward engineering implementation without fully highlighting its academic contribution in relevant research domains.
[1] MindAligner: Explicit Brain Functional Alignment for Cross-Subject Visual Decoding from Limited fMRI Data. ICML 2025

**Questions:**

1. Regarding the dataset split modification in the NSD dataset, could you provide a rigorous justification for this operation and explain how it does not compromise the fairness and generalizability of the experimental results?
2. The MindEye2 baseline results in this paper differ significantly from those in its original paper. Please clarify the reasons for this discrepancy and ensure the fairness of the performance comparison.
3. How does your method compare with existing work like MindAligner in terms of core ideas and performance? Please provide a detailed comparison and explain your method’s unique advantages.
4. Could you elaborate on the research significance of this work, especially from the perspective of biological insights into brain function and neural representation? What new contributions does it make to the field?

---

> ### Author Response · Authors · 2025-11-22
>
> We thank the reviewer for their detailed feedback. We address your concerns below. **W** refers to weakness and **Q** refers to questions. Since there is an overlap in the weaknesses and questions, we address them together:
>
> ---
>
> **W1/Q1:** We thank the reviewer for raising this point. Our modified split is not unfair, and we clarify the motivation below. First, there is no test-set leakage. All methods; including MindEye1 [1], MindEye2 [2], MindBridge [3], and ours are retrained and re-evaluated on the same split, ensuring a fully controlled comparison. Since we re-run every baseline under the new split, the architectural results remain apples-to-apples. Our method does not gain an advantage from the split itself.
>
> Second, the motivation is methodological: Ideal alignment requires a one-to-one mapping between subjects to evaluate alignment quality. Existing NSD splits place the shared images exclusively in the test set, but the best-case scenario cannot be trained or measured if the shared images never appear during training. Moving shared images to the training set is the only way to perform explicit adapter alignment with rigorous supervision in a multi-subject setting.
>
> While semantic alignment can be pursued with similar but non-identical stimuli (please refer to Section 5 where we show that AA does not require exact images), our goal is to quantify the best-case fine-tuning setup for neuroscience practitioners collecting new data (e.g., from lower-resolution/SNR 3T scanners).
>
> Finally, our work does not chase absolute benchmark scores on a predefined NSD split. Our contribution is a training paradigm, not an architectural variant tuned to a particular data partition. All reported gains are relative gains under identical conditions. We hope this addresses your concerns regarding the dataset split. If you have any specific concerns that remain unaddressed we will be glad to answer your queries.
>
> ---
>
> **W2/Q2:** The MindEye2 baseline in our paper is expected to differ from the values reported in the original MindEye2 paper because all methods, including ours, are retrained under a different train/test split, as explained in the previous response. This split is required for explicit cross-subject alignment and makes the evaluation strictly harder (unique images replace shared ones in the test set), which lowers absolute numbers for every baseline, not only MindEye2.
>
> Importantly, the comparison remains fully fair because MindEye1, MindEye2, MindBridge, and our method are all trained under identical data partitions, architectures, and schedules. Our aim is to compare training paradigms and not to reproduce the original MindEye2 benchmark under its original split. Performance gains across different works are relative due to architectural differences and will remain even when testing on different datasets/splits.
>
> We will clarify this in the revision to avoid confusion.
>
> ---
>
> **W3/Q3:** While prior works have explored the general idea of cross-subject alignment, our contributions are distinct in both method and scope.
>
> 1. **Novelty beyond alignment method:** Adapter Alignment (AA) is only one component of our work. We additionally introduce two contributions that are not present in any prior fMRI-to-image system, including MindAligner [4]:
>     - A new greedy image-selection algorithm that identifies the most informative subset of stimuli for low-data fine-tuning.
>     - Transfer experiments demonstrating that AA generalizes to a weaker dataset with a different data resolution (e.g., THINGS), improving cross-dataset and cross-scanner applicability.
>
>     These components are orthogonal to subject alignment and constitute novel contributions on their own.
>
> 2. **Fundamental methodological differences from MindAligner:** MindAligner performs alignment directly in fMRI voxel space, converting one subject’s fMRI pattern into another’s. Its alignment relies on category-level priors, performing a many-to-many mapping because the stimuli differ across subjects. This design is driven by limited shared stimuli, but even with shared stimuli MindAligner still aligns in the raw fMRI domain and still applies category level mapping.
>
>     In contrast, Adapter Alignment operates in the learned common representation space, after the subject-specific adapter (with or without explicit mapping). AA performs a one-to-one mapping between subjects’ common-space embeddings using a two-stage procedure (adapter overfitting + joint fine-tuning). This is architecturally and conceptually different from MindAligner’s end-to-end voxel-space mapping.
>
> 3. **Systematic comparison with related alignment methods:** The methods most comparable to AA in design are works that align at the representation level, such as [3] and [5]. And we compare AA against these works in Appendix F. We test all 3 alignment methods when placed in an identical architectural setting, AA consistently outperforms these prior alignment strategies.

---

> ### Author Response · Authors · 2025-11-22
>
> **W4:**  We thank the reviewer for raising this concern and refer them to our previous responses, where we clarify the evaluation protocol and methodological role of our work. We address the specific points below.
>
> 1. **Our goal is not to surpass MindEye2 or MindAligner benchmarks:** Our paper does not aim to push absolute reconstruction performance. Instead, our contribution is a training paradigm that enables efficient cross-subject modeling with minimal data—an issue repeatedly emphasized in cognitive neuroscience, where collecting 20–40 hours of fMRI per participant at typical rates of \$500-\\$750/hr is infeasible. We intentionally focus on practical data-efficiency, not benchmark chasing. All comparisons in the paper are relative, under identical conditions.
>
> 2. **Scientific significance:** AA is designed to serve as a lightweight cross-subject localizer, analogous to standard neuroscience procedures (e.g., retinotopy, face-localizer scans). A short, shared stimulus set, selected by our greedy algorithm, provides a reusable mapping protocol across participants. This has direct scientific value: it allows labs to reliably build subject-specific models without massive per-subject data collection, expanding the feasibility of fMRI decoding to typical neuroscience settings.
>
> 3. **Biological relevance through representation-level alignment:** Although our main focus is methodological, the work does yield insights into shared representational structure:
>     - Section 2 shows that a subject-agnostic common space emerges naturally during training.
>     - Analysis of singular vectors reveals shared semantic axes (e.g., animate–inanimate, social interaction), consistent with prior findings (Appendix K).
>
> 4. **Relation to MindAligner:** AA and MindAligner operate at different levels of the pipeline and can be synergistic rather than competing. We are currently running additional experiments comparing our method with MindAligner. Initial results show that MindAligner performs comparably to our method and when combined together we get the best possible performance. We will update our response with the averaged results once all the runs complete.
>
> ---
>
> **Q4:** This work represents a methodological step forward that will enable future research in cognitive neuroscience to better understand brain function, as well as extending our understanding of the biological encoding of visual and semantic information in human brains.
>
> First, regarding methodological significance, this work is a critical step towards enabling these natural stimulus reconstruction techniques to be used in typical lab settings where dozens of hours of model-estimation data is not possible per individual participant. Without this ability, we would only be able to achieve insights within the narrow stimulus, task, and population sample constraints of the useful (but limited) NSD study without extensive further funding to acquire large-scale similar datasets across other task regimes. As a result, in our view, this work is highly significant as a methodological advance overcoming a key bottleneck preventing further research using these tools.
>
> Second, regarding biological/neuroscientific significance, this work establishes clear utility of similar high-level representational spaces being implemented in individual brains. That is, this method would not function without a similar joint subspace that is individually instantiated in each participant’s brain. Without implementing any priors for stimulus types, our shared subspace identified principal axes corresponding to major semantic categories, like animate vs inanimate or man made vs natural. Previous studies using targeted stimuli have identified similar stimulus sensitivities, typically organized in spatial patterns [6,7], which have been confirmed in many further NSD analyses and studies on other naturalistic datasets [8,9].
>
> ---
>
> We will add a detailed comparison with MindAligner and update the manuscript as soon as our runs finish. We ask for the reviewer's patience. In the meantime, please let us know if you have any additional questions.

---

> ### Author Response · Authors · 2025-11-22
>
> ### References
>
> 1. Scotti, Paul, et al. "Reconstructing the mind's eye: fmri-to-image with contrastive learning and diffusion priors." Advances in Neural Information Processing Systems 36 (2023): 24705-24728.
>
> 1. Scotti, Paul S., et al. "Mindeye2: Shared-subject models enable fmri-to-image with 1 hour of data." arXiv preprint arXiv:2403.11207 (2024).
>
> 3. Wang, Shizun, et al. "Mindbridge: A cross-subject brain decoding framework." Proceedings of the IEEE/CVF Conference on Computer Vision and Pattern Recognition. 2024.
>
> 4.  Dai, Yuqin, et al. "MindAligner: Explicit Brain Functional Alignment for Cross-Subject Visual Decoding from Limited fMRI Data." arXiv preprint arXiv:2502.05034 (2025).
>
> 5.  Ferrante, Matteo, et al. "Through their eyes: multi-subject Brain Decoding with simple alignment techniques." Imaging Neuroscience 2 (2024): 1-21.
>
> 6. Tripartite organization of the ventral stream by animacy and object size. Talia Konkle, Alfonso Caramazza, Journal Neuroscience 2013
>
> 7. A real-world size organization of object responses in occipitotemporal cortex. Talia Konkle, Aude Oliva, Neuron 2012
>
> 8. A continuous semantic space describes the representation of thousands of object and action categories across the human brain. Huth et al. Neuron 2012
>
> 9. Natural speech reveals the semantic maps that tile human cerebral cortex. Huth et al. Nature 2016

---

### Official Review · Reviewer_V2xv · 2025-10-31

**Soundness:** 4
**Presentation:** 3
**Contribution:** 4
**Rating:** 8
**Confidence:** 5

**Summary:**

This paper well explores data efficiency in fMRI-to-image visual reconstruction with a novel, architecture-agnostic training paradigm called Adapter Alignment (AA). The method first trains a model on a "reference subject" to create a common representation space. New subjects are then aligned to this space using a lightweight adapter before a full end-to-end fine-tuning. This two-stage process, "AAMax," uses an adapter-level MSE loss to maintain alignment. To further boost efficiency, the paper introduces a greedy submodular algorithm for image selection (IS), which picks a small, representative image subset covering the space's principal dimensions. The combined AAMax + IS approach achieves state-of-the-art results in low-data regimes, matching baseline performance with much less data. The method is validated across different architectures (MindEye1, MindEye2), datasets (NSD, THINGS), and varying data sizes.

**Strengths:**

**Originality & Significance:** The paper's primary contribution is significant.
While prior work (e.g., MindEye2) has used common spaces, this paper is the first to argue why this is insufficient and to propose explicitly aligning subjects within that space. The central hypothesis, that subject representations are structurally similar but not semantically aligned, is novel and well-motivated. The "AAMax" two-stage training paradigm is a clever and effective solution. This work directly tackles the main bottleneck in practical fMRI decoding (per-subject data acquisition), offering a solution that could make these powerful models far more accessible to the wider neuroscience community.

**Methodology:** The paper is technically sound and presents a rigorous investigation.

- The initial analysis in Section 2 and Figure 1 provides a clear, data-driven motivation for the entire paper. The use of Procrustes analysis to show structural similarity but misalignment is compelling.

- The proposed Adapter Alignment method is well-explained and the two-stage "AAMax" process is logical.

- The greedy image selection algorithm is a good design, providing a principled, submodular optimization-based alternative to random sampling.

**Experiments:** The experimental validation is comprehensive and convincing.

- The data split methodology is rigorous and ensures fair comparisons.

- The authors correctly identify the method's primary strength in the low-data regime and also completely showing how the performance scales up with data volume (trade-off between data availability and the benefits of AA), clearly demonstrating the capability boundaries of the proposed method.

- The method is shown to be architecture-agnostic.

- The generalization to the THINGS dataset is a strong plus, demonstrating robustness to different acquisition parameters (3T vs. 7T) and stimuli.

- The empirical analyses in Appendix J are excellent, providing direct proof (via eigenvector similarity, MSE, and k-NN consistency) that AAMax does result in a better-aligned common space, which strongly supports the paper's core claim.

**Clarity:** The paper is well-written, and the core concepts are explained clearly. The figures are generally helpful in illustrating the   results.

**Weaknesses:**

**Comparison to Highly Related Work:** The paper fails to discuss the differences with MindBridge (Wang et al., 2024) in the main text or related work (Appendix A, where it's missing). MindBridge's structure is even simpler than MindEye (no diffusion prior), which should be a good architecture for verifying the architecture-agnostic property. It also proposed a cross-subject training algorithm and raised the issue of low data availability for new subjects. Therefore, not discussing this highly related work is a major oversight.

**Figures:**

- Adding a training icon (🔥) would make Figure 2 much easier to understand.

- The paper lacks visual comparison results among different methods.

**Questions:**

The ablation on number of principal eigenvector dimensions is helpful, but why performance is sensitive to this choice?

---

> ### Author Response · Authors · 2025-11-22
>
> We thank the reviewer for their detailed feedback. We are glad you found our work to be well-written and technically sound. We address your concerns below. **W** refers to weakness and **Q** refers to questions:
>
> ---
>
> **W1:** We thank the reviewer for highlighting the relevance of MindBridge [1]. We would like to clarify that MindBridge is discussed in the current submission: specifically in Page 4, Line 208 and Appendix F of the manuscript. There, we describe the differences in alignment strategy and provide a detailed comparison showing that our Adapter Alignment (AA) method offers significantly stronger cross-subject alignment and performance (Appendix F and Table 9).
>
> We agree, however, that MindBridge should also be referenced in the Related Work section. We have added a discussion on MindBridge in our Related Work in the revised manuscript. Due to space constraints, our related work was placed in the appendix, but we will integrate this content into the main text in the camera-ready version.
>
> ----
>
> **W2:** We thank the reviewer for the suggestion. We have updated Figure 2 by adding a clear training icon (🔥) to highlight the alignment step, improving readability and making the figure easier to interpret. The revised figure is in the updated manuscript.
>
> ----
>
> **W3:** Thank you for the suggestion. We have added a visual comparison of reconstructions across MindEye1 [2], MindEye2 [3], and our method (with and without Adapter Alignment). These examples are added in a new appendix (Appendix Q).
>
> ----
>
> **Q1:** The sensitivity to the number of principal eigenvectors arises from how our bin-coverage selection algorithm operates. Each principal direction defines a set of bins, and the greedy set-cover procedure selects images that cover all bins across all dimensions. Increasing the number of included eigenvectors expands the set of constraints the algorithm must satisfy. Each added eigenvector introduces additional bins that must be covered.
>
> - Too few eigenvectors (<15): insufficient coverage of the semantic/structural axes
> - Too many eigenvectors (>40): bins dominated by low-variance directions
>
> Empirically, we observe a stable performance plateau around 20–40 eigenvectors.
>
> ----
>
> We hope we have addressed all your concerns. Please let us know if you have any additional questions
>
> ---
>
> ### References
> 1. Wang, Shizun, et al. "Mindbridge: A cross-subject brain decoding framework." Proceedings of the IEEE/CVF Conference on Computer Vision and Pattern Recognition. 2024.
>
> 1. Scotti, Paul, et al. "Reconstructing the mind's eye: fmri-to-image with contrastive learning and diffusion priors." Advances in Neural Information Processing Systems 36 (2023): 24705-24728.
>
> 1. Scotti, Paul S., et al. "Mindeye2: Shared-subject models enable fmri-to-image with 1 hour of data." arXiv preprint arXiv:2403.11207 (2024).

---

### Author Response · Authors · 2025-11-24
**MindAligner Results**

We thank the reviewers for their patience. To address your requests, we conducted new experiments comparing MindAligner with Adapter Alignment (AA) under the same MindEye1 low-data setup (pretrain on S1, fine-tune on S2 with 1 hour of data using our training split). We evaluated: (1) MindAligner using the exact procedure from the original paper (including categorical pair generation), (2) AA + Image Selection, and (3) a combined pipeline.

AA + Image Selection is comparable with MindAligner despite operating at different points in the pipeline (shared-space vs. voxel-space alignment). Because the two methods address complementary aspects of the alignment problem: MindAligner provides coarse categorical voxel-space alignment, while AA performs fine-grained exact alignment in the shared space, it lets us evaluate a combined 4-stage procedure. The exact details of the training method and setup are detailed in Appendix S of the revised manuscript. This combined approach yields the best overall performance, establishing a new benchmark for MindEye1 in the 1-hour regime. The results are presented below. We will include the comparison with MindAligner in the final version of the manuscript. We hope this addresses the concerns that reviewers **k8hD** and **hpp1** had regarding MindAligner.

---

**Table:** Comparison and combination of MindAligner and Adapter Alignment. All methods use MindEye1 with pretraining on Subject1 and fine-tuning on Subject2 for 1 hour of data. MindAligner is applied exactly as in the original paper using categorical stimulus pairs. AdapterAlignment (AA) uses our 250-image Image Selection subset that spans the breadth of the principal eigenspace. The combined method applies MindAligner for coarse voxel-space alignment followed by AA + Image Selection for fine-grained common-space alignment. The combined approach achieves the strongest overall performance.


| Method                                    | PixCorr ↑ | SSIM ↑ | Alex(2) ↑ | Alex(5) ↑ | Incep ↑ | CLIP ↑ | Eff ↓  | SwAV ↓ |
|-------------------------------------------|-----------|--------|-----------|-----------|---------|---------|--------|--------|
| **MindAligner**                           | *0.116*   | *0.280*| 82.11%    | 88.68%    | 81.36%  | 79.69%  | 0.809  | 0.483  |
| **AdapterAlignment + Image Selection**    | *0.118*   | *0.285*| 81.78%    | 89.61%    | 81.45%  | 79.55%  | 0.810  | 0.491  |
| **MindAligner + AA + Image Selection**    | *0.124*   | *0.284*| **82.24%**    | **89.95%**    | **81.49%**  | **80.99%**  | **0.795** | **0.480**  |

---

### Author Response · Authors · 2025-12-01
**Message to the new area chair**

To the new Area Chair:

We understand the unique circumstances regarding the re-assignment of this paper and the freezing of reviewer scores. We appreciate your time in evaluating our work under these constraints.

Since the original reviewers cannot update their scores or continue the discussion to reflect our rebuttal, we provide this summary to assist you. Our work has three primary contributions:

1. We introduce Adapter Alignment (AA), a training strategy that enables a new subject to be fine-tuned using a reference subject **as long as the two subjects share a small set of semantically similar stimuli**. Prior work either did not align multiple subjects, did not align the full representation pipeline, or did not address data-efficiency. Our experiments show that AA is **architecture-agnostic** and can be incorporated into any existing decoding pipeline, consistently improving fine-tuning performance, especially in the limited-data setting. We also provide direct comparisons showing that AA outperforms or matches related alignment approaches.
2. We present a greedy coverage algorithm that identifies the most informative subset of stimuli by ensuring coverage across the principal eigenspace of the reference subject’s shared representation. This subset is shown to consistently improve alignment and reconstruction. AA + Image Selection (IS) provides the best results in all our experiments.
3. To the best of our knowledge, this is **the first work to demonstrate cross-dataset alignment** and transfer when moving from a high-SNR dataset (NSD 7T) to lower-SNR datasets (THINGS 3T, and now BOLD5K after rebuttal). AA remains effective even without explicit shared stimuli, demonstrating that semantically matched stimuli are sufficient for alignment. We show that fine-tuning a new subject on a strong reference subject is consistently better, even when the two subjects come from different scanners or datasets.

---

We addressed every concern raised by reviewers, either through new experiments, manuscript updates, or detailed rebuttals. Key concerns are summarized below:
- **Comparison with Related Work:** Reviewers *k8hD* and *Hpp1* requested comparisons against MindAligner. We clarified that MindAligner and AA operate at different representational levels: MindAligner performs categorical voxel-space alignment, while AA performs one-to-one alignment in the shared representation space. We provided new experiments (Appendix S) showing that MindAligner and AA+IS operate at different points and can synergize to provide the best results when working together. This also adds comparison with an additional architecture, as requested by reviewer *HkQB*.
- **Soundness of the Dataset Split:** Reviewers *k8hD* and *Hpp1* questioned the modification of the NSD split. We clarified that:
    - The modified split introduces no test leakage.
    - Every baseline was re-trained on the same split, ensuring complete fairness.
    - The new split makes the evaluation strictly harder (unique test images).
    - The absolute baseline numbers naturally differ but all gains are relative within the same split.
- **Need for Explicit Stimuli:** Reviewers **k8hD** and **Hpp1** also suggested our method required identical shared images. We clarified that:
    - The use of explicit shared stimuli in NSD was only to present the ideal alignment scenario.
    - Sec 5 and our new BOLD5K transfer results show that AA works even without shared images and only requires semantically similar stimuli.

We believe the new experimental results and clarifications definitively resolve the technical questions raised. We also provide below a per-reviewer summary of concerns and responses.

**V2xv**
- *Concerns*: missing discussion of a related work and lack of visual comparison.
- *Response:* The former was a misconception which was addressed by pointing to an existing appendix experiment. We further expanded the related work to include this work. The latter was addressed by adding Appendix Q.

**k8hD**
- *Concerns*: dataset split “unfair,” inconsistent MindEye2 baselines, missing MindAligner, and lack of biological insights.
- *Response:* All addressed. Split fairness explained and clarified that baselines were recomputed on new split; MindAligner added; biological insights clarified (Appendix K).

**HkQB**
- *Concerns:* Requested additional dataset transfer and error bars.
- *Response:* Added BOLD5K results in Appendix R and errors bars to subj 2 in Table 2. Will add remaining error bars by the final version.

**Hpp1**
- *Concerns*: Requested MindAligner comparison and clarification of dataset split.
- *Response:* Both fully addressed with new experiments and clarifications. Remaining questions were already addressed in App. F/H (W2), App. C/I (W3) and Sec 5 (W4).

We remain fully available to you during the remainder of the discussion period should you have any specific questions or require further clarification.



Best Regards,

The Authors

---

### Meta-Review · Area_Chair_HE7P · 2026-01-07

**Summary:**

Major issues raised include
1) Lack comparison to highly related work, e.g., MindBridge (Wang et al., 2024); Lack of novelty in core design, comparison with existing public research (e.g., MindAligner[1])

2) Inconsistent baseline results and justification for data split

3) Limited biological insights and conclusions

4) Missing averaged, error bars / confidence intervals

5) More extensive evaluation over datasets and baseline models

**Reviewer Concerns:**

Outstanding issues (using the same numbering as above for easy connection):

(1) Lacking explicit comparison of AdapterAlignment vs MindBridge, both adopt the alignment idea, despite the differences in design/strategy. This is a critical point to be validated but is missing in the table provided, in particular, the results with AdapterAlignment alone is needed. Discussing their difference is not sufficient in this context, as it is a key component of the proposed method as claimed.

(2) The justification of modified data split and fair comparison are still unconvincing:
    * First, the response does not explain why a different data split is needed; Is it because none of existing settings can fit this context? How about the settings used in MindAligner which also considers the cross-subject and low data conditions?
    * Second, retraining and re-evaluating previous methods under the new split setting cannot fully guarantee the fairness as the change in setting may lead to the need for tuning their parameters. Looks like the authors do not carry out this tuning (at least no mention of it).
    * It is unclear what is the definition of "best-case scenario" and why it is required in this context -- looks like new issues introduced.
    * The statement, "Ideal alignment requires a one-to-one mapping between subjects to evaluate alignment quality", is something not explained neither intuitive enough to be understood without elaboration. As it stands, at most, we could consider it as a hypothesis rather than a well established statement or conclusion.

(4) not updated thus far

(5) Added cross-subject and cross-dataset experiments, but lacking comparison with alternatives. Also the analysis is not solid as only one subject from each dataset is used as reference but not multiple.

**Reviewer Scores:**

From the above concerns still outstanding, I do not think the reviewer (e.g., k8hD) would change the score to be positive. AC concur that this work now comes with issues in novelty claim/validation, evaluation setting, and comparison fairness, and is not ready for publication.

---

### Decision · Program_Chairs · 2026-01-26

Reject